# Spliceosome component Usp39 contributes to hepatic lipid homeostasis through the regulation of autophagy

Donghai Cui [1,2,9], Zixiang Wang [1,2,9], Qianli Dang [1,2], Jing Wang [1,2], Junchao Qin[1,2], Jianping Song [3], Xiangyu Zhai[3], Yachao Zhou [1,2], Ling Zhao[1], Gang Lu[4], Hongbin Liu[5], Gang Liu[6], Runping Liu[7], Changshun Shao [8] ✉, Xiyu Zhang [1] ✉ & Zhaojian Liu [1,2,6] ✉

Regulation of alternative splicing (AS) enables a single transcript to yield multiple isoforms that increase transcriptome and proteome diversity. Here, we report that spliceosome component Usp39 plays a role in the regulation of hepatocyte lipid homeostasis. We demonstrate that Usp39 expression is downregulated in hepatic tissues of non-alcoholic fatty liver disease (NAFLD) and non-alcoholic steatohepatitis (NASH) subjects. Hepatocyte-specific *Usp39* deletion in mice leads to increased lipid accumulation, spontaneous steatosis and impaired autophagy. Combined analysis of RNA immunoprecipitation (RIP-seq) and bulk RNA sequencing (RNA-seq) data reveals that Usp39 regulates AS of several autophagy-related genes. In particular, deletion of Usp39 results in alternative 5' splice site selection of exon 6 in Heat shock transcription factor 1 (*Hsf1*) and consequently its reduced expression. Importantly, overexpression of *Hsf1* could attenuate lipid accumulation caused by Usp39 deficiency. Taken together, our findings indicate that Usp39-mediated AS is required for sustaining autophagy and lipid homeostasis in the liver.

Non-alcoholic fatty liver disease (NAFLD) is the most common liver disease, with a global prevalence of 25%[1], and is strongly associated with metabolic syndrome, obesity, and diabetes[2]. Non-alcoholic steatohepatitis (NASH) is a more severe subtype of NAFLD characterized by diffuse fatty infiltration and inflammation[3]. NASH can lead to fibrosis, cirrhosis and hepatocellular carcinoma[4], and is now the second leading cause of liver transplantation. Currently, no drugs have been approved by the FDA for the treatment of NAFLD and NASH, and the pathogenesis of NAFLD is not clearly understood.

In NAFLD, hepatic fat accumulation is the result of an imbalance between lipid acquisition and disposal, and recent evidence suggests that autophagy is an important modulator of hepatic metabolism and that defective autophagy may contribute to the pathogenesis of NAFLD[5]. Autophagy mediates the breakdown of intracellular lipid droplets in hepatocytes through the process of lipophagy[6]. Hepatic loss of *Atg7* increases lipid accumulation due to defective autophagy, leading to NAFLD in mice[7], and *ATG7* gene mutations in human patients increase the risk of NAFLD[8]. In addition, liver-specific knock-out of *Tfeb* promotes high-fat diet (HFD) induced hepatic steatosis[9].

[1]Key Laboratory of Experimental Teratology, Ministry of Education, School of Basic Medical Science, Department of Obstetrics and Gynecology, Qilu Hospital, Shandong University, Jinan, China. [2]Advanced Medical Research Institute, Shandong University, Jinan, China. [3]Department of General Surgery, The Second Hospital, Shandong University, Jinan, China. [4]CUHK-SDU Joint Laboratory on Reproductive Genetics, School of Biomedical Sciences, The Chinese University of Hong Kong, Hong Kong, China. [5]Center for Reproductive Medicine, Shandong University, Jinan, China. [6]Nephrology Research Institute of Shandong University, The Second Hospital of Shandong University, Jinan, China. [7]School of Chinese Materia Medica, Beijing University of Chinese Medicine, Beijing, China. [8]Institutes for Translational Medicine, State Key Laboratory of Radiation Medicine and Protection, Soochow University, Suzhou, China. [9]These authors contributed equally: Donghai Cui, Zixiang Wang. ✉e-mail: shaoc@suda.edu.cn; xiyuzhang@sdu.edu.cn; liujian9782@sdu.edu.cn

Conversely, hepatocyte-specific *Rubicon* knockout effectively inhibits HFD-induced hepatic steatosis in mice[10]. The involvement of defective autophagy in NAFLD suggests that augmentation of autophagy may ameliorate fatty liver disease.

Usp39 (Inactive Ubiquitin-Specific Peptidase 39) is also called U4/U6. U5 tri-snRNP associated protein 2, and shares 65% amino acid identity with yeast Sad1[11]. Usp39 is not required for the stability of the U4/U6. U5 tri-snRNP, which is the largest pre-assembled spliceosomal complex containing snRNAs and proteins[12]. but is essential for recruitment of the tri-snRNP to the pre-spliceosome[13]. Mounting evidence suggests that Usp39 is involved in the development of various types of cancer[14,15]. *Usp39* deficient mice display early embryonic lethality with aberrant apico-basal polarity[16], but the physiological and pathological functions of Usp39 in liver remain largely unknown.

Here, we demonstrate an important physiological role of Usp39 in liver function. We observed that Usp39 was significantly downregulated in livers of human NASH patients and mice with NAFLD/NASH. Our study also showed that *Usp39* deficiency in hepatocytes causes autophagy defects and lipid accumulation and thus results in spontaneous steatosis in mice. Mechanistically, Usp39 was found to regulate AS of autophagy-related genes, including Heat Shock Transcription Factor (*Hsf1*), a master transcriptional factor involved in development, metabolism and aging[17]. *Usp39* deficiency results in an isoform of *Hsf1* that undergoes degradation by nonsense-mediated mRNA decay, which in turn leads to impaired autophagy and to hepatic steatosis. Our findings therefore demonstrate that by sustaining the expression of genes required for execution of autophagy through AS, Usp39 maintains hepatic lipid homeostasis.

## Results

### Hepatic Usp39 expression is decreased in NAFLD and NASH

To identify the components of the U4/U6. U5 tri-snRNP complex involved in the development of fatty liver disease, we analyzed the mRNA levels of genes encoding 16 core U4/U6. U5 tri-snRNP components using transcriptomic data in the livers of chow-fed and HFD-fed mice from GEO database (GSE165855)[18], and another cohort of chow-fed and CDAHFD-fed mice from GEO database (GSE154892)[19]. Among them, Usp39 and Ddx23 were significantly downregulated in RNA-seq data of both HFD and NASH mice (Fig. 1a, b). We next performed interaction network analysis of 16 U4/U6. U5 tri-snRNP components using the two RNA-seq data sets and found *Usp39* to be one of the hub factors that interact most strongly with the U5 snRNP and U4/U6 snRNP components in the liver of NAFLD and NASH mice (Fig. 1c). Interestingly, *USP39* expression was negative correlated with NAFLD activity score in a cohort (GSE193084)[20] of human NAFLD patients (Fig. 1d; Supplementary Table 1), we then focused on USP39 for further investigation. We analyzed the expression level of *USP39* in the cohort of human NAFLD patients and found *USP39* to be significantly downregulated in high NAFLD activity score group compared with low NAFLD activity score group (Fig. 1e). Consistently, *USP39* level was lower in late fibrosis stage (2–4) than in early fibrosis stage (0–1) (Fig. 1f and Supplementary Table 1). Decreased *Usp39* expression was also detected in transcriptomic data of NAFLD and NASH mouse model (Fig. S1a, b). We next measured its expression in the livers of HFD-fed and methionine-choline-deficient (MCD)-fed mice compared to those of chow-fed control mice by immunoblotting and qPCR. The expression level of Usp39 was significantly lower in the livers of HFD-fed and MCD-fed mice compared to those of chow-fed control mice (Fig. 1g, h and Fig. S1c–f). Importantly, USP39 expression was also found to be decreased in the livers of human patients with NASH compared to those without NASH (Fig. 1i). It should be noted that there was a lower band (around 55 kDa) in addition to the full-length band (around 65 kDa) (Fig. 1g–i). This extra band was present when Usp39 was examined by using three different antibodies (Fig. S1e). It may arise as a splicing variant, but how it exactly originates and functions remains to

be determined. Further subcellular localization analysis confirmed that Usp39 was localized in the nuclei (Fig. S1g, h). We also performed immunohistochemistry staining and found that Usp39 was mainly localized in the nuclei, and its protein level was dramatically decreased in HFD-fed and MCD-fed mice (Fig. 1j and Fig. S1i). It was co-localized with nuclear speckle marker SC35 as evidenced by immuno-fluorescence assay (Fig. S1j). Collectively, the reduced expression of Usp39 in NAFLD and NASH liver implicates Usp39 in the progression of these diseases.

### Hepatic *Usp39* deletion impairs liver development and homeostasis

To explore the role of Usp39 in liver development and homeostasis, we measured Usp39 expression by qPCR and immunoblotting and found the expression level of Usp39 to be gradually reduced in the mouse livers with increasing age (Fig. S2a–c). We then generated hepatic-specific *Usp39* knockout mice (*Usp39*-HKO) by crossing *Usp39*[fl/fl] mice to *Alb-Cre* mice (Fig. 2a, b) and Fig. S2d–h). *Usp39*-HKO mice exhibited significantly reduced body weight between 4 and 8 weeks after birth (Fig. 2c, d and Fig. S2i). At the age of 5 weeks, the body weight, liver weight, and liver/body weight ratio were also reduced in *Usp39*-HKO mice (Fig. 2e–g). H&E staining revealed central vein fibrosis and reduced numbers of binuclear cells in *Usp39*-HKO livers compared to their littermate controls (Fig. 2h left). We also observed an increased expression of proliferative markers (*Ki67* and *Pcna*) hepatocyte proliferation in *Usp39*-HKO livers compared with control (Fig. 2h–j and Fig. S2j). Of note, mRNA expression of *Afp* and *H19* (progenitor cell markers) was significantly increased, whereas albumin expression was decreased (Fig. 2k–m). We have also measured several markers (ALT, AST, LDH, albumin levels and immune cell infiltration) of liver function. The results showed that at the age of 5 weeks, *Usp39* deletion induced spontaneous liver injury, as revealed by increased levels of ALT, AST, LDH and a decreased level of Alb (Fig. 2n). Depletion of *Usp39* induced liver fibrosis as assessed by Sirius red staining and qPCR analysis of fibrosis markers (Fig. 2o, p). No apoptotic cells, inflammatory cells and progenitor cell activation were found in *Usp39*-HKO mice at this age (Fig. 2o, p and Fig. S3a–f). The results showed that *Usp39* deletion induced spontaneous liver injury at the age of 5 weeks. Interestingly, liver injury was significantly alleviated in 10-week-old *Usp39*-HKO mice compared to 5-week-old *Usp39*-HKO mice (Fig. 2n–p and Fig. S3g–l). These data suggest the essential role of Usp39 in early postnatal liver development in mice.

### Hepatocyte-specific *Usp39* deletion induces spontaneous steatosis

To determine the specific role of hepatic Usp39 in liver homeostasis and function, we examined *Usp39*-HKO and control mice at 12-month-old. No significant differences in body weight or liver weight were found between *Usp39*-HKO mice and control mice (Fig. S4a–d). Intriguingly, the spleens were enlarged and livers pale-colored in *Usp39*-HKO mice (Fig. 3a), and there were massive accumulation of large lipid droplets and fibrosis as evidenced by H&E and Sirius Red staining, respectively (Fig. 3b). Consistently, hepatic TG levels were higher *Usp39*-HKO mice than in controls (Fig. 3c). Furthermore, inflammatory and fibrotic genes (*Col1a1, α-Sma, Tnf-α*, and *Tgfb1*) were generally upregulated (Fig. 3d). We also examined livers of *Usp39*-HKO and control mice at 5 weeks of age. *Usp39*-HKO livers were pale in comparison to the reddish-brown appearance in healthy control, particularly after fasting (Fig. S4e). Oil Red O staining of liver tissue sections revealed a remarkably increased lipid accumulation in *Usp39*-HKO mice, which was more pronounced after fasting (Fig. S4f). In parallel, fasting induced greater hepatic triglyceride (TG) accumulation in *Usp39*-HKO mice than in control mice (Fig. S4g), whereas free fatty acid levels in both the liver and serum were similarly elevated in *Usp39*-HKO mice and control mice upon fasting (Fig. S4h and S4i), suggesting that fatty acids secretion and uptake are not affected by hepatic deletion of

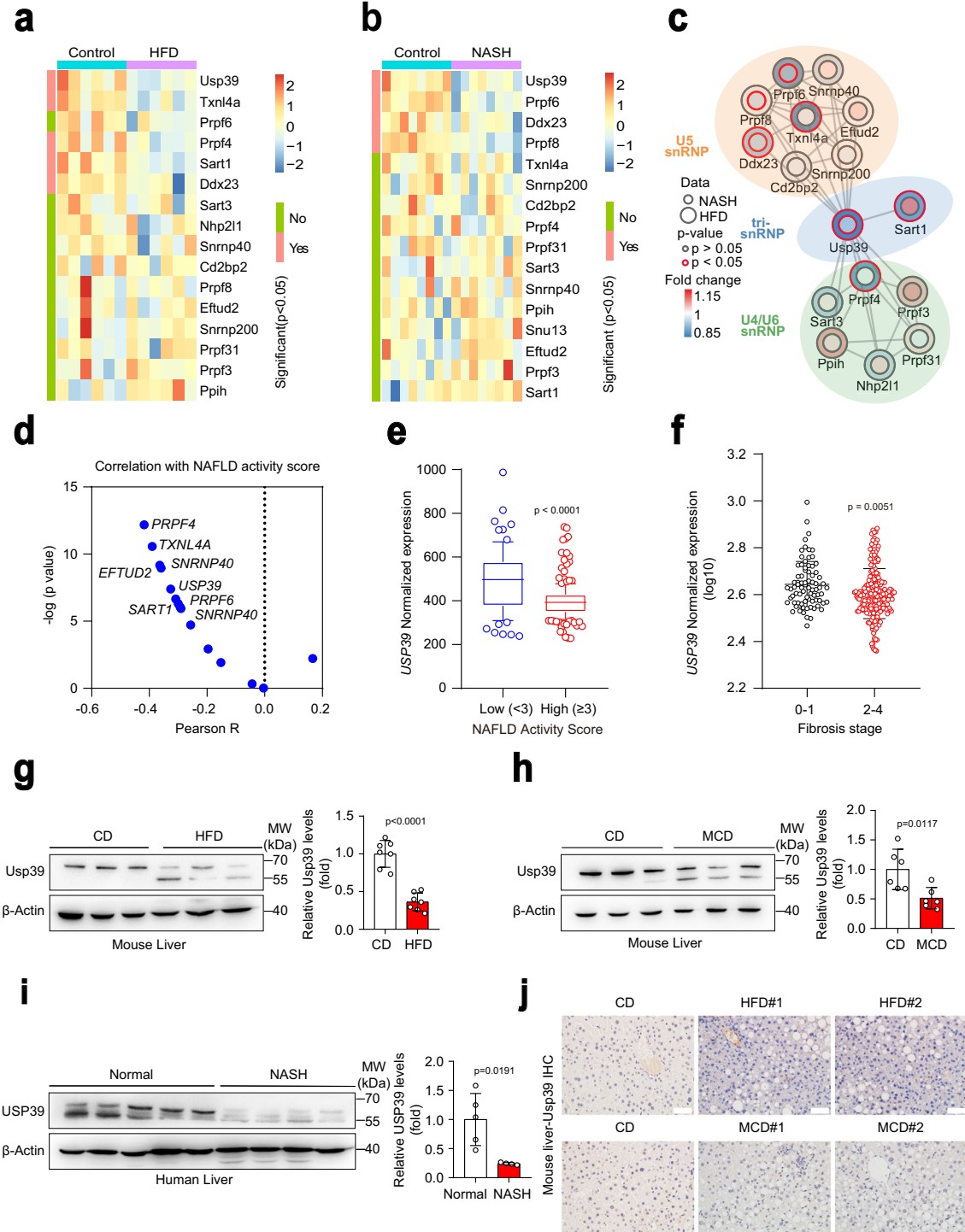

**Fig. 1 | Hepatic Usp39 expression is decreased in NAFLD and NASH. a, b** Heatmap showed the expression of core U4/U6. U5 tri-snRNP components in the mouse liver models of NAFLD ($n = 6$) and NASH ($n = 8$) compared with controls using publicly available RNA-seq data (GSE165855 and (GSE154892). The quantification data was shown in Supplementary Fig. S1a, b. **c** Interaction network analysis of 16 U4/U6. U5 tri-snRNP components using RNA-seq data (GSE165855) and (GSE154892) visualized with Cytoscape. U4/U6. U5 complex related splicing factors were classified into U5 snRNP (orange), U4/U6 snRNP (green), and tri-snRNP (blue) according to Spliceosome DB[42]. **d** Correlation analysis of U4/U6. U5 splicing factors expression and NALFD activity score based on RNA-seq data of a human NAFLD cohort of 271 patients. **e** *USP39* mRNA expression was compared between high (score ≥ 3) ($n = 196$) and low (score <3) ($n = 75$) NALFD activity score group. The bounds of the box were the upper and lower quartile with the median value in the center. The whiskers indicated the minima and maxima. The range from 10–90 percentile. **f** *USP39* mRNA expression was compared between stage 0–1 ($n = 78$) and stage 2–4 ($n = 192$) fibrosis stage group. **g, h** Immunoblotting was performed to measure Usp39 protein in the livers of HFD-fed ($n = 7$ per group) and MCD-fed ($n = 6$ per group) male mice and chow-fed mice. **i** The USP39 protein level was measured in human healthy ($n = 5$) and NASH ($n = 4$) livers. **j** Immunohistochemical (IHC) staining of Usp39 in the liver sections of MCD-fed and HFD-fed male mice compared to those of the chow-fed mice ($n = 12$ per group). Scale bars, 50 μm. IHC image were quantified by pathologists. Resultes were shown in Fig. S1i. All immunoblotting band intensities were quantified by Image J. The *p*-value was obtained by Pearson R Correlation (**d**) and others performed unpaired two-sided Student's *t*-test, and the results are presented as the mean ± S.D. CD, control diet. MW, molecular weight. Source data are provided as a Source data file.

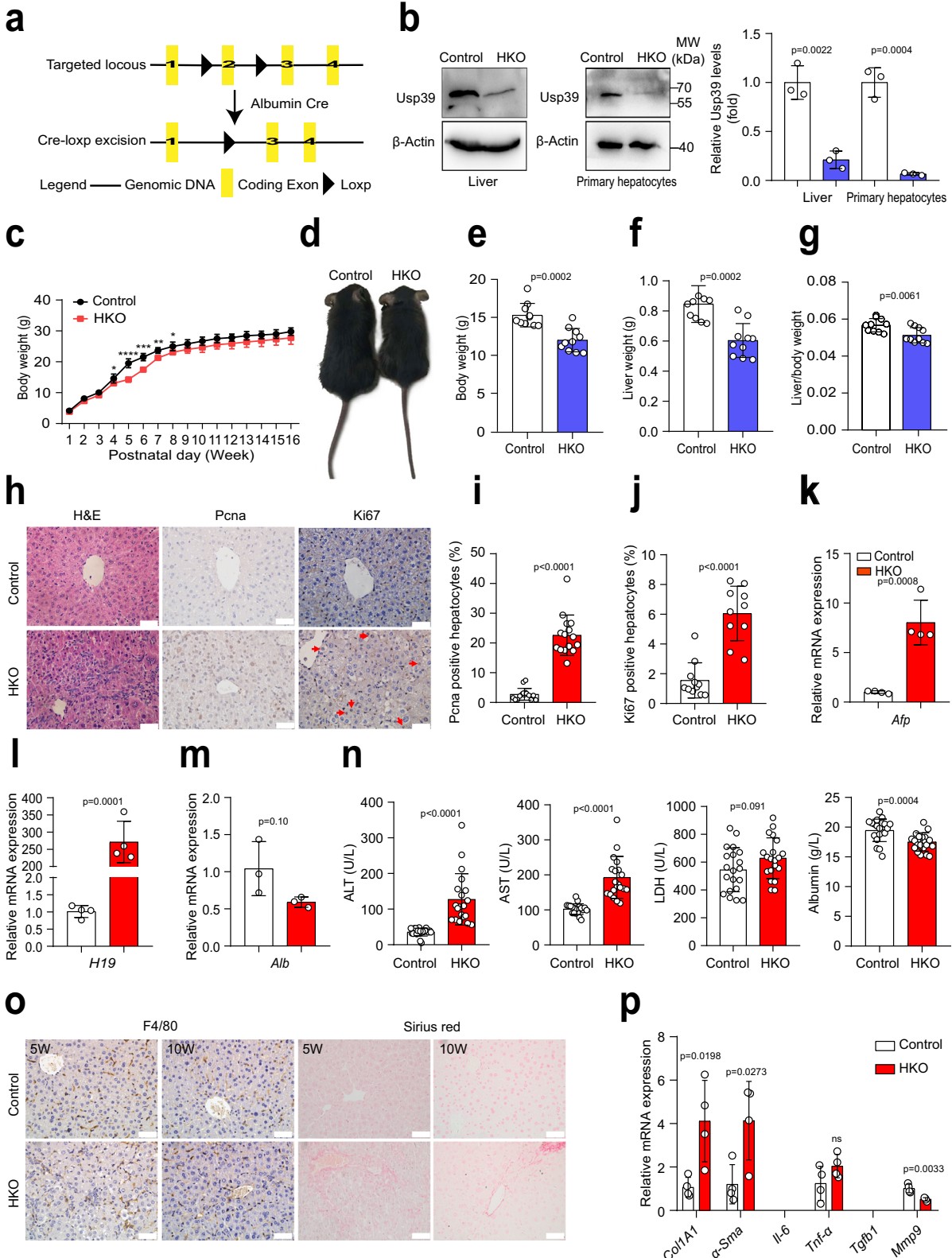

*Usp39*. Consistently, Oil Red O staining of 10-week-old mouse liver tissue sections confirmed hepatic lipid accumulation upon hepatocyte-specific *Usp39* deletion (Fig. S4j–n). These findings suggest that hepatocyte-specific knockout of *Usp39* led to lipid accumulation and spontaneous steatosis in mice.

To further delineate the role of Usp39 in hepatic steatosis, *Usp39*-HKO and control mice were challenged with 12 weeks of HFD feeding.

*Usp39*-HKO mice exhibited more pronounced hepatic steatosis and fibrotic formation with HFD feeding (Fig. 3e, f). The hepatic TG level and the mRNA expression of inflammatory and fibrotic genes were also more dramatically increased (Fig. 3g, h). Furthermore, *Usp39*-HKO and control mice were treated with an MCD diet for 5 weeks. Consistent with HFD model, hepatic-specific *Usp39* deletion exacerbated hepatic steatosis induced by the MCD diets (Fig. S4o–r). Moreover, the liver

**Fig. 2 | Hepatic *Usp39* knockout impairs liver homeostasis and function.**
**a** Schematic illustrating the generation of the *Usp39* conditional knockout mice. The strategy was to insert loxp sites to flank exons 2 of the mouse *Usp39* gene. *Usp39*$^{fl/fl}$ mice were crossed with albumin-Cre mice to generate *Usp39*-HKO mice. **b** Immunoblotting was performed to determine *Usp39* knockout efficiency using liver tissues and primary hepatocytes ($n = 3$ independent experiments). Band intensities were quantified by Image J. **c** Body weight of male mice was recorded from 1 to 16 weeks after birth ($n = 5$ per group). **d** Representative images of 5-week-old control and *Usp39*-HKO male mice. **e**–**g** Body weight, liver weight, and liver/body weight ratio were compared between control and *Usp39*-HKO mice ($n = 10$ per group). **h** H&E staining ($n = 6$), IHC staining of Pcna ($n = 15$) and Ki67 ($n = 10$) in the liver sections of 5-week-old male mice. Red arrows indicate Ki67-positive nuclei. Scale bars, 50 μm. **i** The Pcna-positive cells count of 5-week-old male mice liver sections ($n = 15$). **j** The Ki67-positive cells count of 5-week-old control ($n = 11$) and *Usp39*-HKO ($n = 10$) male mice liver sections. **k**–**m** Relative mRNA expression *Afp* ($n = 4$), *H19* ($n = 4$), and *Alb* ($n = 3$) in the liver of 5-week-old male mice. **n** Serum ALT, AST, LDH and albumin levels were measured in 5-week-old male mice fasted for 16 h ($n = 20$ per group). **o** IHC staining of F4/80, Sirius Red staining in the liver sections of 5- and 10-week-old male mice ($n = 5$ per group). Scale bars, 50 μm. **p** qPCR was performed to analyze mRNA expression indicated genes in livers of 5-week-old male mice ($n = 4$ per group). Images are representative of at least three independent experiments. The *p*-value was obtained by unpaired two-sided Student's *t*-test, and the results are presented as the mean ± S.D. *$p < 0.05$, **$p < 0.01$, ***$p < 0.001$, ****$p < 0.0001$. NS, stands for non-significant. *Alb*, albumin. *Afp*, α-Fetoprotein. MW, molecular weight. Source data are provided as a Source data file.

lipidomes and transcriptomes from 5-week-old *Usp39*-HKO and control mice ($n = 6$) after fasting were characterized by LC/MS and RNA-seq, respectively. A total of 1692 lipid species across 31 lipid classes in liver samples were identified (Fig. S4s, t), we found that TG, diglyceride (DG), phosphatidylinositol (PIP2) and sphingoid base (So) were significantly increased in *Usp39*-HKO mice compared to control mice (Fig. 3i, j). A combined lipidome and transcriptome analysis showed that liver fibrosis and steatosis were highly enriched in *Usp39*-HKO mice (Fig. 3k–n). Taken together, these results suggest that *Usp39*-HKO mice are more susceptible to fatty liver disease.

## Autophagy is inhibited in mice with hepatocyte-specific *Usp39* deletion

Excessive hepatic lipid accumulation results from an imbalance between lipid acquisition and lipid disposal. To elucidate the mechanism by which *Usp39* deletion leads to increased lipid accumulation, we first measured genes related de novo lipogenesis (*Srebp-1c*, *Chrebp-α*, *Chrebp-β*, *Acc1*, *Scd1*, *Elovl6*, *Fasn*, and *Pklr*) and lipid lipolysis (*Atgl*, *Hsl*, *Lpl*, and *Lipa*). Strikingly, lipogenesis-related genes were significantly downregulated (Fig. S5a) and lipolysis-related genes were upregulated (Fig. S5b) in *Usp39*-HKO livers compared with controls. In addition, no significant differences in lipid secretion were found between *Usp39*-HKO and control mice, as evidenced by injected poloxamer 407 (inhibitor of peripheral lipid uptake), followed by TG measurement in *Usp39*-HKO and control mice (Fig. S5c). Furthermore, metabolic assessment revealed that 5-week-old *Usp39*-HKO mice consumed less food than control mice. No difference in food intake was observed between 10-week-old *Usp39*-HKO and control mice. In addition, no significant difference in Energy Expenditure, RER was found between 5- and 10-week-old control and *Usp39*-HKO mice (Fig. S5d, e). These results suggest that the increased lipid accumulation in *Usp39*-HKO livers was unlikely to result from defective lipolysis, de novo lipogenesis, increased food intake, or lower metabolic rate.

Autophagy plays a key role in the degradation of lipid droplets in the liver[21], and our Ingenuity Pathway Analysis of lipidome and transcriptome data showed that autophagy, TG degradation and Fatty Acid β-oxidation (FAO) were pathways significantly diminished, while the mTOR signaling pathway was activated in *Usp39*-HKO livers compared with controls (Fig. 4a). We hypothesized that *Usp39* depletion might block autophagy and thereby increase lipid droplet accumulation in liver. To explore this, we assessed the expression of autophagy-related genes by qPCR and found that *Ulk1*, *Atg3* and *Atg7* were significantly downregulated in *Usp39*-HKO livers compared with controls (Fig. 4b). Consistently, the mRNA levels of *Ulk1*, *Atg3* and *Atg7* were decreased in AML12 cells (a mouse hepatocyte cell line) and primary hepatocytes upon *Usp39* knockdown (Fig. 4c and Fig. S5f). Furthermore, immunoblotting revealed that Ulk1, Atg7 and LC3 II were decreased and p62 was increased upon *Usp39* knockout or knockdown (Fig. 4d–f and Fig. S5g–m). We have also measured *p62* and *LC3B* mRNA level by RT-qPCR and but observed no significant difference between control and *Usp39*-HKO livers (Fig. S5n, o). Decreased formation of autophagic vesicles in *Usp39*-HKO livers was evident under electron microscopy (Fig. 4g). Moreover, co-immunofluorescence staining of Plin2 and Lamp2 revealed an increased abundance of Plin2 and a reduced distribution of Lamp2 (Fig. 4h). Consistently, EGFP-LC3 puncta were significantly decreased in in primary hepatocytes and AML12 cells upon *Usp39* depletion (Fig. 4i and Fig. S6a). Importantly, knockdown of *Usp39* in primary cells and AML12 cells decreased the number of EGFP-LC3 puncta that co-localized with BODIPY $C_{12}$ stained lipids (Fig. 4j and Fig. S6b). Fatty acid oxidation (FAO) related genes assessed by qPCR were all found to be significantly downregulated in *Usp39*-HKO livers compared with controls (Fig. S6c). Moreover, oxygen consumption rates (OCR) measured by Seahorse in control and oleic acid-treated primary hepatocytes and AML12 cells were significantly decreased in *Usp39* knockdown cells (Fig. S6d, e). Taken together, these results indicate that the lipid accumulation in *Usp39*-HKO livers might occur due to impaired autophagy.

## Hepatocyte-specific *Usp39* deletion leads to aberrant splicing of autophagy-related genes

Given that Usp39 is a component of U4/U6. U5 tri-snRNP, we hypothesized that loss of Usp39 might disrupt RNA splicing of genes required for hepatic lipid autophagy. To this end, we performed RNA-seq on liver tissues of *Usp39*-HKO and control mice ($n = 4$). The rMATs software was used for AS events analysis and a total of 1993 splicing events in 1472 genes were detected. The predominant AS events upon *Usp39* deletion were exon skipping (55%), alternative 5' splice sites (12%) and intron retention (12%) (Fig. 5a). Heatmap showed the differentially expressed autophagy-related genes in liver tissues of *Usp39*-HKO mice compared to control mice (Fig. 5b), and a volcano plot was constructed to show the distribution of differential AS events upon *Usp39* deficiency (Fig. 5c). To identify the genome-wide binding sites and targets of Usp39, we conducted RNA immunoprecipitation sequencing (RIP-seq) in AML12 cells. Ultrasound fragmentation of RNA and stringent washing conditions improved specificity and resolution of binding signals. RIP peak distribution analysis revealed that Usp39 was predominantly mapped to introns (29%), exons (18%) and 3'UTRs (19%) (Fig. 5d). Motif enrichment analysis of the RIP-seq data identified GGCCAC and GCCAUA as the two most abundant elements (Fig. 5e and Fig. S7a–f). Further analysis revealed that Usp39 binding Motif 1 were clearly enriched near the splice sites of autophagy genes (Fig. S7d–f). To identify candidate AS targets regulated by Usp39, we integrated Usp39-bound genes and AS- related genes using RIP-seq and RNA-seq data and identified 611 common genes (Fig. 5f). Gene Ontology analysis of the 611 genes showed that the pathways related to the regulation of cellular catabolic processes, spliceosome complex assembly, peptidyl-serine phosphorylation and autophagy were significantly enriched (Fig. 5g). In order to identify functional targets regulated by Usp39 in hepatic lipid accumulation, we examined the AS patterns of autophagy-related genes that were enriched in Usp39-bound and alternatively spliced transcripts including *Ulk3*, *Nrbp2*, *Tcirg1*, *Trp53inp1*, and *Hsf1*. We then validated the

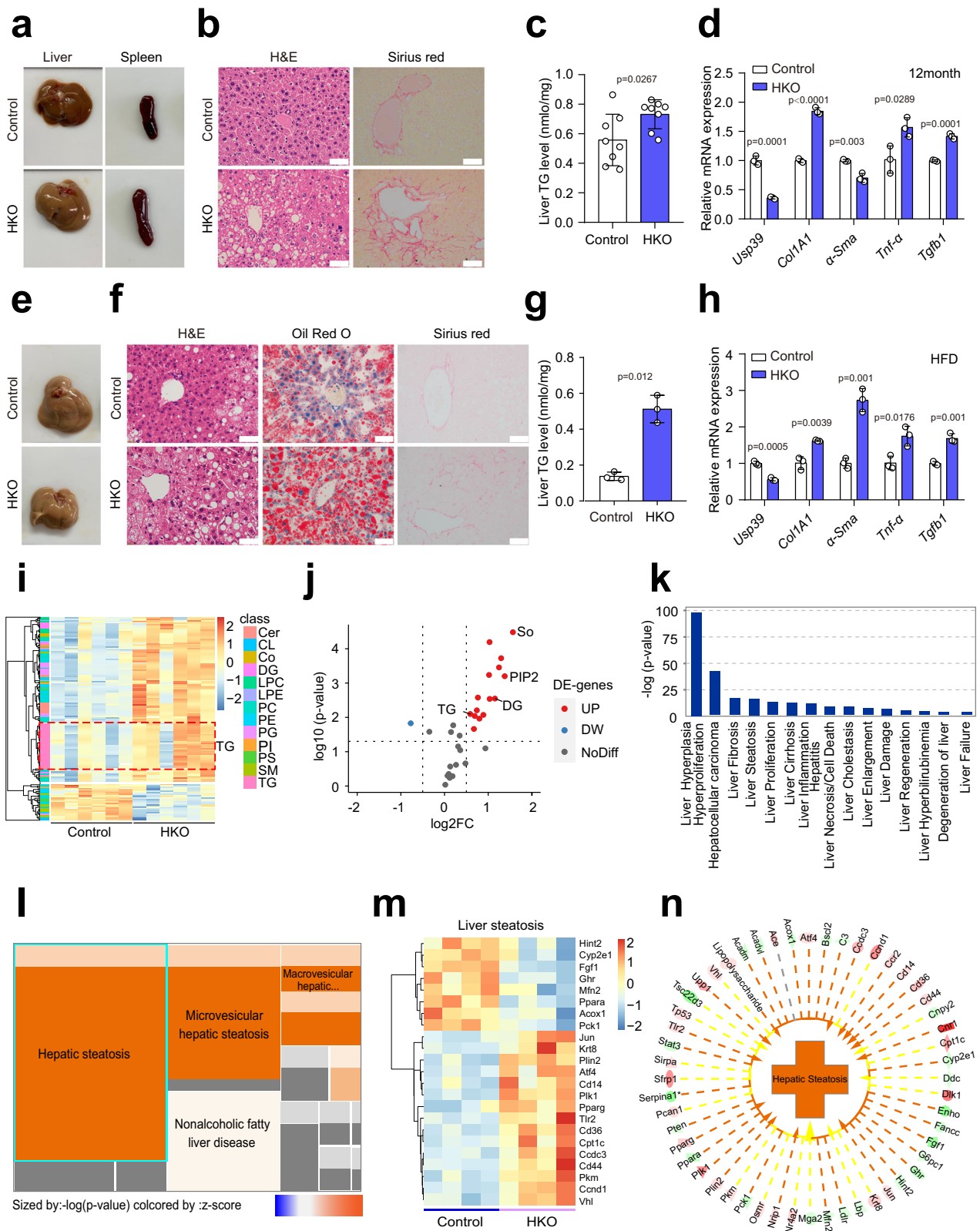

altered splicing of these genes by AS and binding peak analysis (Fig. 5h–k) as well as by RT-PCR (Fig. 5l).

### Usp39 deletion leads to mis-splicing and degradation of Hsf1 in hepatocytes

We next focused on Hsf1, the master transcriptional regulator of heat shock response[22], that promotes cytoprotective autophagy[18,23]. We

analyzed the splicing events of Hsf1 and found that disruption of Usp39 led to alternative 5' splice site selection of exon 6, which introduces a premature termination codon (PTC) that may result in nonsense-mediated mRNA decay (NMD) (Fig. 6a, b). Indeed, the altered splicing of Hsf1 was validated in Usp39-HKO livers by semi-quantitative RT-PCR (Fig. 6c). A minigene splicing assay further confirmed the altered splicing outcome of Hsf1 decreased with increasing addition of Usp39,

**Fig. 3 | Spontaneous steatosis in hepatocyte-specific *Usp39* knockout mice.**
**a** Representative liver and spleen images from 12-month-old male mice fasted for
16 h. **b** H&E staining and Sirius Red staining of liver sections in 12-month-old male
mice (*n* = 3 per group). Scale bars, 50 μm. **c** Hepatic TG levels were analyzed in 12-
month-old male mice fasted for 16 h (*n* = 8 per group). **d** qPCR was performed to
analyze mRNA expression of indicated genes in livers of 12-month-old male mice
(*n* = 3). **e** Representative liver images from control and *Usp39*-HKO male mice fed
the HFD for 12 weeks. **f** H&E (left), Oil Red O (middle), and Sirius Red staining (right)
of liver sections from 16-week-old male mice fed the HFD for 12 weeks (*n* = 3). Scale
bars, 50 μm. **g** Hepatic TG levels were analyzed in16-week-old male mice fed the
HFD for 12 weeks (*n* = 3). **h** qPCR was performed to analyze mRNA expression of
indicated genes in livers of control and *Usp39*-HKO male mice fed the HFD for
12 weeks (*n* = 3). **i** Heatmap showing lipidomic data in the livers of 5-week-old male

mice fasted 16 h (*n* = 6). **j** Volcano plot of the differential lipids between the livers
(*n* = 6 per group). **k** Pathway enrichment analysis using Ingenuity Pathway Analysis
(IPA) of the lipidomic and RNA-seq data. **l** Enrichment level of the subcategories of
liver steatosis shown in (**k**). Gray squares indicate the activate patterns of the
subcategories are not available in reference. **m** Heatmap showing differential
hepatic steatosis-related genes in *Usp39*-HKO mice compared to control mice (*n* = 4
per group). **n** IPA analysis of hepatic steatosis-related genes using RNA-seq data in
control and *Usp39*-HKO mice. Correlation of differential genes with hepatic stea-
tosis were shown with arrows (activation), and dashes (inhibition). Images are
representative of at least three independent experiments. The *p*-value was obtained
by IPA (**k**–**l**) and others performed unpaired two-sided Student's *t*-test, and the
results are presented as the mean ± S.D. NS, stands for non-significant. Source data
are provided as a Source data file.

but increased when Usp39 was depleted (Fig. 6d and Fig. S8b). Sub-
sequently, we analyzed the canonical isoform and NMD isoform of *Hsf1*
by qPCR in liver tissues and found that canonical isoform was sig-
nificantly decreased while the NMD isoform was increased upon loss of
*Usp39* (Fig. 6e). The canonical isoform was also significantly decreased
while the NMD isoform was increased upon loss of Usp39 in HFD-fed
and MCD-fed mice compared to those of the chow-fed mice (Fig.
s8c–f). Splicing regulation of *HSF1* by USP39 was further determined in
Human cell lines Huh7 and HepG2. The results showed that canonical
isoform was significantly decreased while the NMD isoform was
increased upon knockdown of *USP39* (Fig. S8g–j). To verify that NMD
isoform is degraded by NMD pathway, we measured the RNA half-life
of NMD isoform in *Upf1* knockdown and control AML12 cells treated
with actinomycin D. As expected, the half-life of NMD isoform of *Hsf1*
was significantly increased in *Upf1* knockdown cells relative to controls
(Fig. 6f). To provide further evidence for the direct binding of Usp39 to
*Hsf1*, we performed a RIP assay in AML12 cells overexpressing FLAG-
Usp39. RIP-qPCR demonstrated that Usp39 bound to four sites around
exon 6 of *Hsf1* (Fig. 6g), and the RNA pull-down assay further revealed
successful pull-down of Usp39 by a biotin-labeled Hsf1 probe (Fig. 6h).
Consequently, mRNA expression of *Hsf1* and autophagy-related genes
was significantly reduced in *Usp39*-HKO livers compared with controls
(Fig. S8k), and in AML12 cells with *Usp39* knockdown (Fig. S8l). In
contrast, the expression of *Hsf1* and autophagy-related genes was
significantly increased in AML12 cells overexpressing Usp39 (Fig. S8m).
We further found that the protein levels of Hsf1, Ulk1, Atg7, and the LC3
II were significantly decreased and p62 was increased upon *Usp39*
depletion under fed (Fig. 6i and Fig. S9a), fasted (Fig. 6j and Fig. S9b)
and HFD (Fig. 6k and Fig. S9c) conditions. A tamoxifen-inducible
Usp39 knockout mouse model (*Usp39*^fl/fl^; *UBC-Cre*^ERT2^) further con-
firmed the decreased level of Hsf1 upon *Usp39* deletion (Fig. S9d, e).
The protein level of HSF1 was similarly decreased in human cell lines
Huh7 and HepG2 upon *USP39* knockdown (Fig. S9f). Moreover, we
observed decreased protein level of Hsf1 in the livers of HFD-fed and
MCD-fed mice compared to those of the chow-fed mice (Fig. 6l, m and
Fig. S9g, h). Finally, decreased HSF1 expression was also detected in
the livers of patients with NASH compared with those without NASH
(Fig. 6n and Fig. S9i). Taken together, these findings indicate that
deletion of *Usp39* results in altered splicing and rapid degradation
of *Hsf1*.

## Hsf1 promotes autophagy and alleviates hepatic steatosis caused by *Usp39* depletion
We next performed rescue experiments to determine whether the
downregulation of *Hsf1* mediates the lipid accumulation resulting from
loss of *Usp39*. BODIPY staining in AML12 cells showed that lipid accu-
mulation caused by *Usp39* deletion could be rescued by forced Hsf1
expression (Fig. 7a, b). In contrast, knockdown of *Hsf1* by siRNA
impaired lipid degradation triggered by *Usp39* overexpression
(Fig. S10a, b). Immunofluorescence of Plin2 and Lamp2 staining further
revealed that impaired autophagy induced by *Usp39* deletion was

restored by forced Hsf1 expression reversed in AML12 cells (Fig. 7c).
Conversely, silencing of Hsf1 in Usp39 overexpressing AML12 cells had
the opposite effect (Fig. S10c). Immunoblotting showed that the
impaired autophagy by *Usp39* deletion could be rescued by forced
Hsf1 expression in AML12 cells, particularly under serum starvation
(Fig. 7d and Fig. S10d). In contrast, the enhanced autophagy caused by
overexpression of Usp39 could be abrogated by *Hsf1* knockdown (Fig.
S10e, f). To strengthen the conclusion that Hsf1 is a functional target of
Usp39 and promotes autophagy in hepatic lipid metabolism, we
injected Usp39 floxed mice with AAV8-TBG-Cre or AAV-TBG-Null to
generate liver-specific *Usp39*-HKO mice. While there was no difference
in body weights, liver weight and liver/body weight ratio (Fig. S10g),
the livers were pale and the hepatic TG level was significantly increased
in AAV-TBG-Cre mice (Fig. S10h, i). These mice were then infected with
adenovirus expressing Hsf1 or empty vector, followed by 4 weeks of
normal chow diet. Ectopic Hsf1 expression clearly alleviated the
hepatic steatosis induced by *Usp39* deletion (Fig. 7e). Consistently,
impaired autophagy caused by *Usp39* deletion was restored, as evi-
denced by immunostaining of Plin2 and Lamp2 (Fig. 7f) and immu-
noblotting of autophagy-related genes (Fig. 7g and Fig. S10j).
Collectively, these results suggest that hepatic steatosis due to *Usp39*
depletion is partially mediated by Hsf1 downregulation and the sub-
sequent impairment in autophagy.

## Discussion
We here demonstrate a physiological role for Usp39 in the main-
tenance of liver lipid homeostasis. Our results show that hepatocyte-
specific *Usp39* deletion in mice leads to spontaneous hepatic steatosis.
*Usp39* depletion blocks autophagy and thereby increases lipid droplet
accumulation in the liver. Mechanistically, *Usp39* deficiency was shown
to result in mis-splicing and downregulation of autophagy-related
genes including Hsf1. The finding that Usp39-mediated AS promotes
autophagic activity to maintain hepatic lipid homeostasis is consistent
with the recognition that autophagy accounts for a high percentage of
lipolysis in the liver[21].

As a component of the U4/U6. U5 tri-snRNP complex, Usp39 is
crucial for pre-mRNA splicing[24]. Because of the early embryonic leth-
ality in constitutive *Usp39* knockout mice, the physiological roles of
Usp39 have not been fully elucidated. A recent study reported the role
of Usp39 in B cell development using B cell-specific *Usp39* conditional
knockout mice and showed that Usp39 is essential for B cell devel-
opment in the bone marrow, although RNA splicing is barely affected
in *Usp39*-deficient B cells[25]. Nevertheless, a previous study demon-
strated that Usp39 is essential for the recruitment of the tri-snRNP to
the pre-spliceosome and for the splicing function[26]. In addition, Usp39
has been shown to be involved in splicing of Aurora B which is essential
for proper spindle checkpoint function[11]. Our study provides strong
evidence that Usp39 plays an important role in splicing regulation.
Using RIP-seq and RNA-seq approaches, we found *Usp39* depletion
leads to 1993 altered AS events, with the predominant AS events being
exon skipping (55%), alternative 5' splice sites (12%), and intron

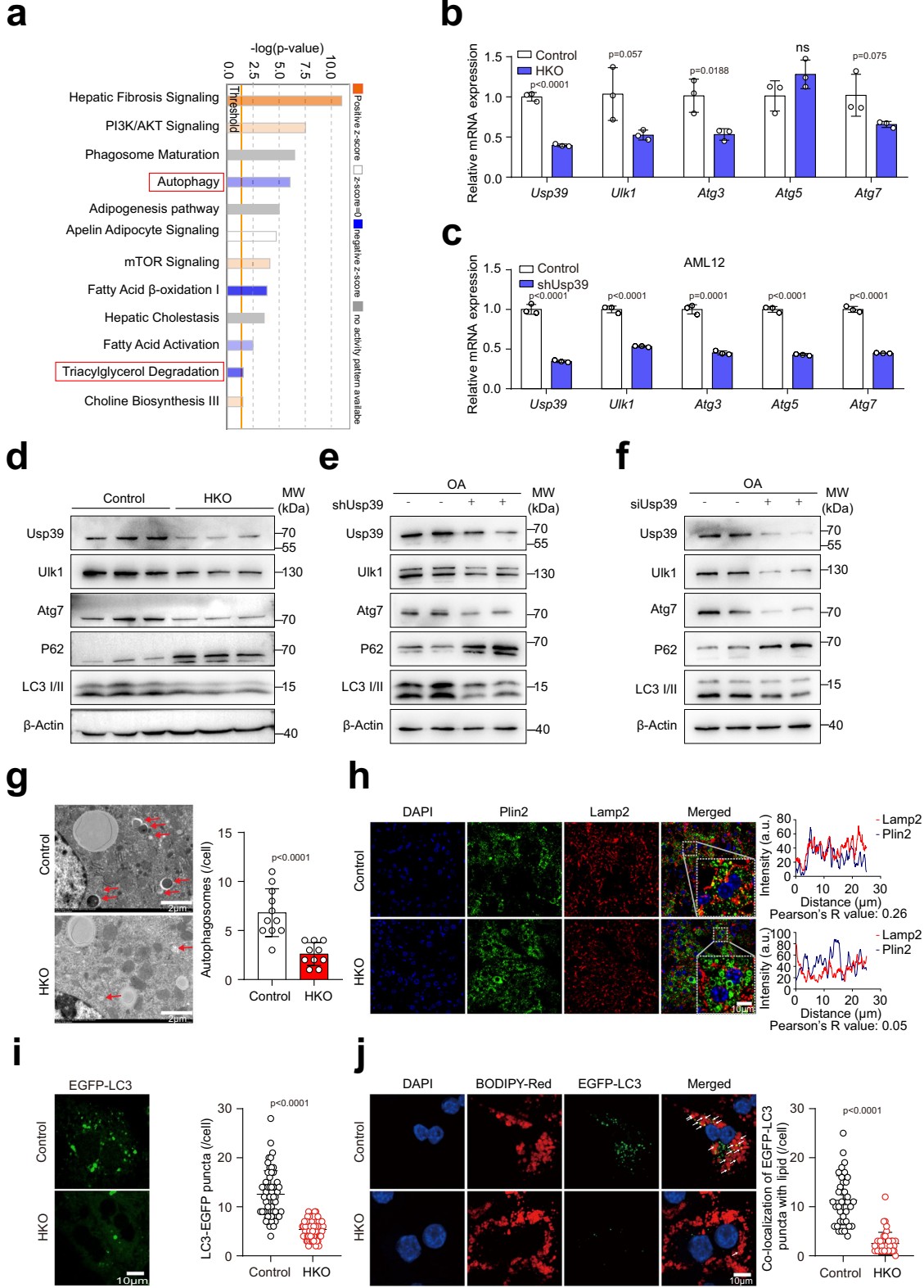

retention (12%) (Fig. 5a). We further integrated Usp39-bound genes with AS related genes using RIP-seq and RNA-seq data and identified 611 Usp39-regulated AS candidate target genes including *Hsf1*. Our findings suggest that Usp39 is a major splicing factor that regulates multiple splicing events in hepatic lipid metabolism.

One of the key findings from our combined analysis of RIP-seq and RNA-seq data is that Usp39 regulates AS of several autophagy-related genes including *Hsf1*, which has been well documented to play a role in

heat shock response by preventing protein aggregation and proteome imbalance[22]. Hsf1 has also been reported to be involved in non-stressed circumstances, including development, metabolism and aging[17], and Hsf1 activation protects against HFD-induced insulin resistance and steatosis[27] while overexpression of Hsf1 induces autophagy to improve survival and proteostasis in *Caenorhabditis elegans*[28]. In addition, Hsf1 activates autophagy through transcriptional regulation of autophagy-related genes including *Atg5*, *Atg7*, and *Atg12*[18,29]. Hsf1 expression and

**Fig. 4 | Autophagy is impaired in hepatocytes lacking *Usp39*. a** Pathway enrichment analysis of differentially expressed genes using RNA-seq data for control and *Usp39*-HKO male mice. **b** qPCR analysis of autophagy-related genes in the livers (*n* = 3 per group). **c** qPCR was performed to analyze expression of autophagy-related genes in *Usp39* knockdown and control AML12 cells (*n* = 3). **d** Immunoblotting was performed to analyze protein levels of autophagy-related genes in 5-week-old control and *Usp39*-HKO male mice (*n* = 3 per group) fasted for 16 h. **e, f** Immunoblotting was performed to analyze protein levels of autophagy-related genes in *Usp39* knockdown and control AML12 cells (*n* = 3) (**e**) and primary cells (*n* = 4) (**f**) supplemented with 0.4 mM oleic acid for 6 h. Immunoblotting band intensities were quantified by Image J, which were shown in Fig. S5g–i. **g** Electron micrographs in the livers of control (*n* = 11) and *Usp39*-HKO (*n* = 10) male mice showing autophagic vesicles with quantification. Scale bars, 2 μm. **h** Confocal image of the liver sections in control and *Usp39*-HKO mice showing co-localization of Plin2 (green) and Lamp2 (red). Scale bars, 10 μm. **i** Confocal images of primary hepatocytes transfected with Ad-EGFP-LC3 and control vectors followed by serum deprivation to 2 h, showing EGFP-LC3 puncta/cell (*n* = 56 cells for control, *n* = 64 cells for HKO). Scale bars, 10 μm. **j** Co-localization of EGFP-LC3 puncta and BODIPY C$_{12}$ (white arrows) in primary hepatocytes transfected with Ad-EGFP-LC3 and control vectors for 72 h, cells were then cultured in medium supplemented with 0.4 mM oleic acid (*n* = 37 cells for control, *n* = 39 cells for HKO). Scale bars, 10 μm. Immunofluorescence staining and co-localization was quantified by Image J. Images are representative of at least three independent experiments. The *p*-value was obtained by IPA (**a**) and others performed unpaired two-sided Student's *t*-test, and the results are presented as the mean ± S.D. NS, stands for non-significant. OA, Oleic acid. MW, molecular weight. Source data are provided as a Source data file.

activity is regulated at multiple levels, including protein-protein interactions and posttranslational modifications[17]. For example, Sirt1 deacetylates Hsf1 and upregulates its binding to the Hsp70 promoter[30]. Our data revealed that Usp39 regulates the AS of *Hsf1*, and *Usp39* deficiency leads to altered splicing of exon 6, resulting in an isoform that harbors a premature termination codon that leads to its degradation by nonsense-mediated mRNA decay. Indeed, the protein level of Hsf1 was significantly decreased upon *Usp39* deletion. We compared the sequences of human and mouse Hsf1 gene near the splice site of exon 6 and found them to be highly conserved and contain two Usp39 binding motif 1 (Fig. S8a). In line with our findings, the stress-sensitive splicing factor SF3B1 has been shown to regulate both HSF1 concentration and activity[31]. Taken together, these findings suggest that Usp39-mediated AS regulates *Hsf1* expression.

Our study further demonstrated that impaired autophagy caused by *Usp39* deficiency plays a key role in hepatic lipid accumulation. The liver is rich in lysosomes and highly active in autophagy, a feature critical to basic liver functions[5]. Defective autophagy is implicated in the development and progression of NAFLD[32], and autophagy serves as a major regulator of lipid homeostasis in the liver, by degrading lipid droplets via lysosomes[33]. Pharmacological inhibition of autophagy or liver-specific knockout of *Atg7* blocks autophagy and promotes hepatic lipid accumulation[21]. Our data revealed that autophagy-regulated genes including *Ulk1*, *Atg3* and *Atg7*, and *Hsf1* were significantly downregulated upon *Usp39* depletion. Importantly, our results suggest that loss of *Usp39* induces hepatic steatosis partially by impairing Hsf1-regulated autophagy. Considering that hepatic lipid accumulation in *Usp39*-HKO livers might also be caused by other mechanisms, we analyzed the expression level of genes related to lipolysis, β-oxidation, and de novo lipogenesis in *Usp39*-HKO livers compared with controls. Our results revealed that hepatic lipid accumulation in *Usp39*-HKO livers was not due to defective lipolysis, β-oxidation or increased de novo lipogenesis. Taken together, these data suggest that the hepatic lipid accumulation induced by *Usp39* depletion is largely mediated by an impaired autophagy.

In summary, our findings demonstrate a role for the spliceosome component Usp39 in the regulation of autophagy and hepatocyte lipid homeostasis. *Usp39* deficiency in hepatocytes causes autophagy defects and lipid accumulation and thus leads to spontaneous steatosis. Mechanistically, Usp39 regulates the AS of autophagy-related genes, including *Hsf1*, to promote autophagy. Collectively, our findings of AS in regulating autophagy and hepatocyte lipid homeostasis make it a potential target for treating fatty liver diseases.

## Methods
### Mice
*Usp39*[fl/+] and Albumin-Cre mice on C57BL/6J background were obtained from GemPharmatech Laboratory (China), and *UBC*-Cre[ERT2] mice on C57BL/6J background were obtained from the Jackson Laboratory. The *Usp39*-floxed mouse line was generated by flanking exon 2 of *Usp39* with *loxP* sites. To generate hepatic-specific *Usp39*-HKO mice, *Usp39*[fl/fl] mice were crossed to the Albumin-Cre mouse line. For diet-induced model, control and *Usp39*-HKO male mice were fed CD or 60% kcal fat HFD (D12492, Research Diets) at 4 weeks of age for 12 weeks. For MCD (Methionine and Choline Deficient L-Amino Acid Diet) induced model, 10-week-old male mice were fed an MCD (XSYT-ED-009, Biotechnology) for 5 weeks. For the in vivo rescue experiments, AAV-empty, AAV-TBG-Cre, or Ad-Hsf1 virus were injected into 8-week-old male *Usp39*[fl/fl] mice followed by a normal chow diet for 4 weeks. The male mice were then subjected to 16 h fasting prior to sacrifice. Adenovirus and adeno-associated virus were purchased from Jinan WZ Biosciences INC. All animals used in our study were housed under a controlled lighting regime (12 h light; 12 h darkness) at 21–22 °C and 60% ± 10% relative humidity with freely available food and water. All animal experiments were conducted in accordance with the guidelines of the Animal Care and Use Committee at Shandong University (ECSBMSSDU2022-2-113). Primers for PCR genotyping are listed in Supplementary Table 2.

### Cell lines and constructs
The AML12 (mouse liver cell line) (CRL-2254, USA) and HepG2 (HB-8065, USA) was from the ATCC, and Hepa1-6 was a generous gift from Prof Peihui Wang, Advanced Medical Research Institute, Shandong University. HEK293T (SCSP-502, China) and Huh-7 (SCSP-526, China) cell line was obtained from the National Collection of Authenticated Cell Cultures. AML12 cells were cultured in DMEM-F12 medium with ITS liquid medium supplement (Sigma), and 40 ng/mL dexamethasone, while the Hepa1-6, Huh-7, HepG2 and 293T cell lines were cultured in DMEM medium. The PLKO.1-shUsp39-TRC vector was obtained from WZ Biosciences INC. The pT3G-3FPS-Usp39 and pLenti-Hsf1-C-Myc-DDK-IRES-Puro lentiviral plasmids were constructed by colony PCR. AML12 cells stably expressing Usp39 and Hsf1 were obtained by viral infection with pT3G-3FPS-Usp39, Lenti-Hsf1-c-myc-DDK-IRES-puro, shUsp39-AML12 cells were constructs by viral infection with PLKO.1-shUsp39-TRC. All virus-infected cells were cultured in complete medium containing 2 μg/mL puromycin. Transient transfections of AML12, Hepa1-6, HepG2, Huh7 and primary hepatocyte cells were transfected with plasmids (1 μg) or siRNAs (50 nM) for 48 or 72 h. Cells were then harvested for the following experiments. The siRNA, shRNA sequences and primer sequences are shown in Supplementary Table 2. All mediums were supplemented with 10% fetal bovine serum (Gibco) and 100 units/mL penicillin-streptomycin solution. Cells were cultured in a humidified atmosphere of 5% $CO_2$ at 37 °C.

### Isolation and culture of primary mouse hepatocytes
Primary mouse hepatocytes were isolated using a two-step collagenase digestion method as previously described[34]. Five-week-old male *Usp39*-HKO and control mice were used for primary isolation. Briefly, after the tissues were digested via a perfusion of collagenase type IV solution. The isolated cells were resuspended in 50 ml of medium, filtered with double-layer sterile gauze, centrifuged at

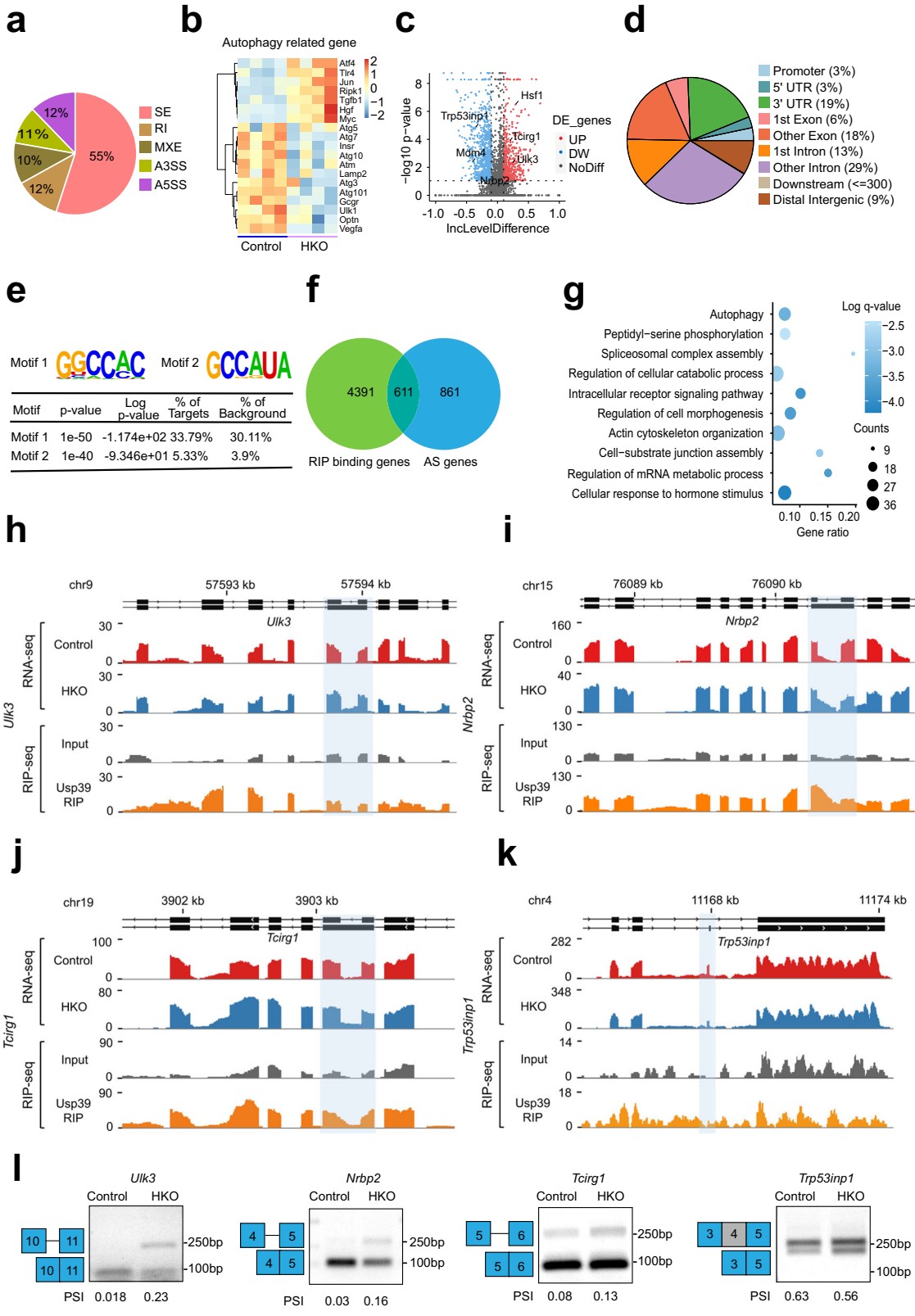

60 × *g* for 5 min, and the procedure was repeated. The cells were finally resuspended in 10 ml medium. Cell yield and viability were scored by trypan blue staining. Cells were plated and cultured in Dulbecco's Modified Eagles Medium (DMEM) medium containing 10% fetal bovine serum (Gibco), 100 units/mL penicillin-streptomycin solution, 1 μM T4 thyroid hormone, 10 μM dexamethasone Metone and 1.5 μM insulin (Macgene, CC101).

### RNA extraction and PCR

Total RNA was isolated from mouse liver or mouse cell lines using TRIzol reagent according to the manufacturer's protocol (Invitrogen, 15596-026). A total of 1 μg of total RNA was reverse-transcribed into cDNA using the HiScript III RT SuperMix for qPCR (+gDNA wiper) (Vazyme, R223-01) according to the manufacturer's instructions. Real-time PCR was performed using ChamQ SYBR Colour qPCR Master Mix

**Fig. 5 | *Usp39* deletion leads to altered splicing of autophagy-related genes in hepatocytes. a** Pie chart showing the distribution of AS events in the RNA-seq data in the livers of 5-week-old control and *Usp39*-HKO male mice (*n* = 4 per group) fasted 16 h. A total of 1937 splicing events from 1472 genes were identified. SE, skipped exons; RI, retained introns; A5SS, alternative 5′ splice site; A3SS, alternative 3′ splice site; MXE, mutually exclusive exons (*p* < 0.05, |delInclevel| > 0.1). **b** Heatmap of autophagy-related genes using RNA-seq data of control and *Usp39*-HKO male mice (*n* = 4 per group). **c** Volcano plot showing differential AS events in the RNA-seq data of livers from *Usp39*-HKO male mice compared to controls. Negative log10 transformed *p*-values and the delInclevel values were analyzed with rMATS (*n* = 4 per group). **d** The distribution of Usp39 binding peaks (35, 980), corresponding to 5002 genes, were identified from RIP-seq data with Piranha software (-p 0.001 -z 100). **e** HOMER de novo motif analysis of Usp39 binding peaks based on the RIP-seq data showing top-two Usp39 binding motifs. Binding motifs were ranked by *p*-value, and the statistical results including the percentage of targets and, the background are shown. **f** Venn diagram of 611 overlap genes from 5002 Usp39 binding genes (RIP-seq) and 1472 differentially expressed genes (RNA-seq). **g** Biological process enrichment analysis of the 611 genes from (**f**). Accumulative hypergeometric q-values and gene ratios were calculated. **h**–**k** Sashimi plots of AS and binding peaks for indicated genes using RNA-seq and RIP-seq data. The light blue region highlights the AS sites and the Usp39 binding region. **l** PCR was performed to analyze AS events in indicated genes using splice-variant specific primers in the liver tissues of 5-week-old control and *Usp39*-HKO male mice. Images are representative of at least three independent experiments. The results are presented as the mean ± S.D. NS, stands for non-significant. RIP, RNA immunoprecipitation sequencing. AS, alternative splicing. Source data are provided as a Source data file.

(Vazyme, Q411-02) on a Quant Studio 3 thermocycler (Thermo Fisher). Semi-quantitative real-time PCR was used to analyze AS products. Primer sequences are listed in Supplementary Table 2.

## Histological analysis

After the mice were scarified, left lateral lobe of the liver was removed and fixed in 4% neutral paraformaldehyde and embedded in paraffin. Paraffin-embedded or frozen sections of liver tissue sections (4–6 µm) were stained with hematoxylin and eosin stain (H&E), immunohistochemical staining, or immunofluorescence staining[35]. For immunohistochemical experiments, Paraffin sections were deparaffinized in xylene and rehydrated with a graded series of ethanol solutions. Antigen retrieval was performed in EDTA by heating in a microwave. Tissue slides were blocked with 5% BSA and incubated with primary antibodies against rabbit anti-Usp39 (1:100 Abcam ab131332), Ki67 (1:50 Abcam ab15580), Pcna (1:500 Servicebio GB11010-1), F4/80 (1:500 Servicebio GB11027) overnight at 4 °C. Bright field images were obtained using an Olympus BX53 microscope system. The IHC staining score for each sample was determined by two pathologists in a blinded manner based on the intensity and extent of staining across the section. The intensity of staining was scored as 0 (negative), 1 (weak), 2 (moderate), and 3 (strong). The extent of staining was based on the percentage of positive Usp39 cells: 1 (0−25%), 2 (26−50%), 3 (51−75%), and 4 (76−100%). The final IHC staining score was generated by multiplying the percentage score with the staining intensity score. The final score was the average of two pathologists. The primary antibodies are listed in Key resources Supplementary Table 3.

## Immunofluorescence staining

Cell slides or ice sections of liver tissue were fixed in 5% paraformaldehyde followed by permeabilization with 0.5% Triton X-100. Samples were then blocked with 5% BSA for 1 h and incubated with primary antibodies against rabbit anti-Usp39 (1:400 Abcam ab131244), Sox9 (1:200 Millipore AB5535), Albumin (1:400 Proteintech 16475-1-AP), Perilipin 2 (1:200 Proteintech 15294-1-AP), rat anti-Lamp2 (1:200 Abcam ab13524), mouse anti-SC35 (1:400 Abcam ab11826). Fluorescent images were captured on a Dragonfly 200 confocal microscopy system (Andor Technology). Immunofluorescence staining and co-localization was quantified by Image J. The primary antibodies are listed in Key resources Supplementary Table 3.

## BODIPY and Oil Red O staining

Primary hepatocytes from C57BL/6 WT male mice and AML12 cells were seeded in 24-well plates 6 h before transfection. Primary hepatocyte and AML12 cells were infected with Usp39 siRNA or shUsp39 vector and the equal number of AD-EGFP-LC3 adenoviruses overnight. After 48 h, 0.4 mM OA and 0.5 µM BODIPY $C_{12}$ treatment for 12 h and serum starvation treatment cells for 3 h. Living cell fluorescent images were taken on a Dragonfly 200 confocal microscopy system (Andor Technology). For Oil Red O staining, the frozen sections of liver tissue were placed at room temperature for 30 min. Slides were stained with prepared 0.3% Oil Red O solution (Merck, O625) for 15 min and then counterstained with hematoxylin.

## Mouse serum assay and liver function measurements

The blood was collected in EP tubes in a total volume of 1 ml. The serum was separated by centrifugation and transferred to new EP tubes for storage at −80 degrees. Serum alanine aminotransferase (ALT), aspartate aminotransferase (AST), albumin (albumin) and lactate dehydrogenase (LDH) activities were measured using kits from Sigma according to the manufacture's protocols.

Triglyceride (TG) secretion was measured in 5-week-old male mice injected intraperitoneally with 0.25 g/kg of poloxamer 407[36]. The retro-orbital venous blood was collected from the mice at 0 min and 300 min after injection. The serum TG content was measured using triglyceride detection kit. For liver TG and non-esterified fatty acid (NEFA) measurement, 30–40 mg of liver tissues were homogenized in 400 µl of deionized water and lipids were extracted with 1 ml of a 2:1 chloroform: methanol. The mixture was incubated for 4 h at room temperature with vigorous shaking for complete extraction. The mixture was then centrifuged at 13,400 × *g* for 5 min to allow the liquid surface to delaminate. The upper chloroform phase was transferred to a new EP tube. It was dried under a stream of air and then regenerated by adding 200 µL of anhydrous ethanol and mixing in a water bath at 42 °C for 5 min. TG and NEFA were measured using colorimetric kits according to the manufacture's protocols.

## Metabolic assessment in mice

Mice body weight was measured weekly during the experiment. Columbus Instruments Comprehensive Lab Animal Monitoring System CLAMS6 was used to measure the metabolic rate of 5- and 10-week-old *Usp39*-HKO and Control male mice (*n* = 5–7). Mice were adapted to individual metabolic chambers for three days prior to data collection. Oxygen consumption, RER (breathing entropy), energy expenditure, food intake, and locomotor activity were recorded every seven minutes for over 48 h. Data were generated by automatic recording of the instrument. These data were analyzed at 12 h dark :12 h light cycle at 22–23 °C.

## Mitochondrial respiration measurement

Primary hepatocytes and AML12 cells were seeded in 96 well Seahorse plates at $1 \times 10^4$ cells per well. Prior to mitochondrial respiration measurement, medium was replaced with XF Assay Medium (pH 7.4, supplemented with 10 mM glucose, 1 mM sodium pyruvate, and 2 mM glutamine) and incubated for 1 h at 37 °C in the absence of $CO_2$. Cellular OCR were recorded at baseline and then the cells were challenged injected with 1.5 µM oligomycin, 2 µM FCCP, 0.5 µM rotenone, 0.5 µM antimycin. After 4 h, OCR were measured again. The OCR values were normalized to the protein levels of each well. The Seahorse Wave software was used to analyze the data (Agilent).

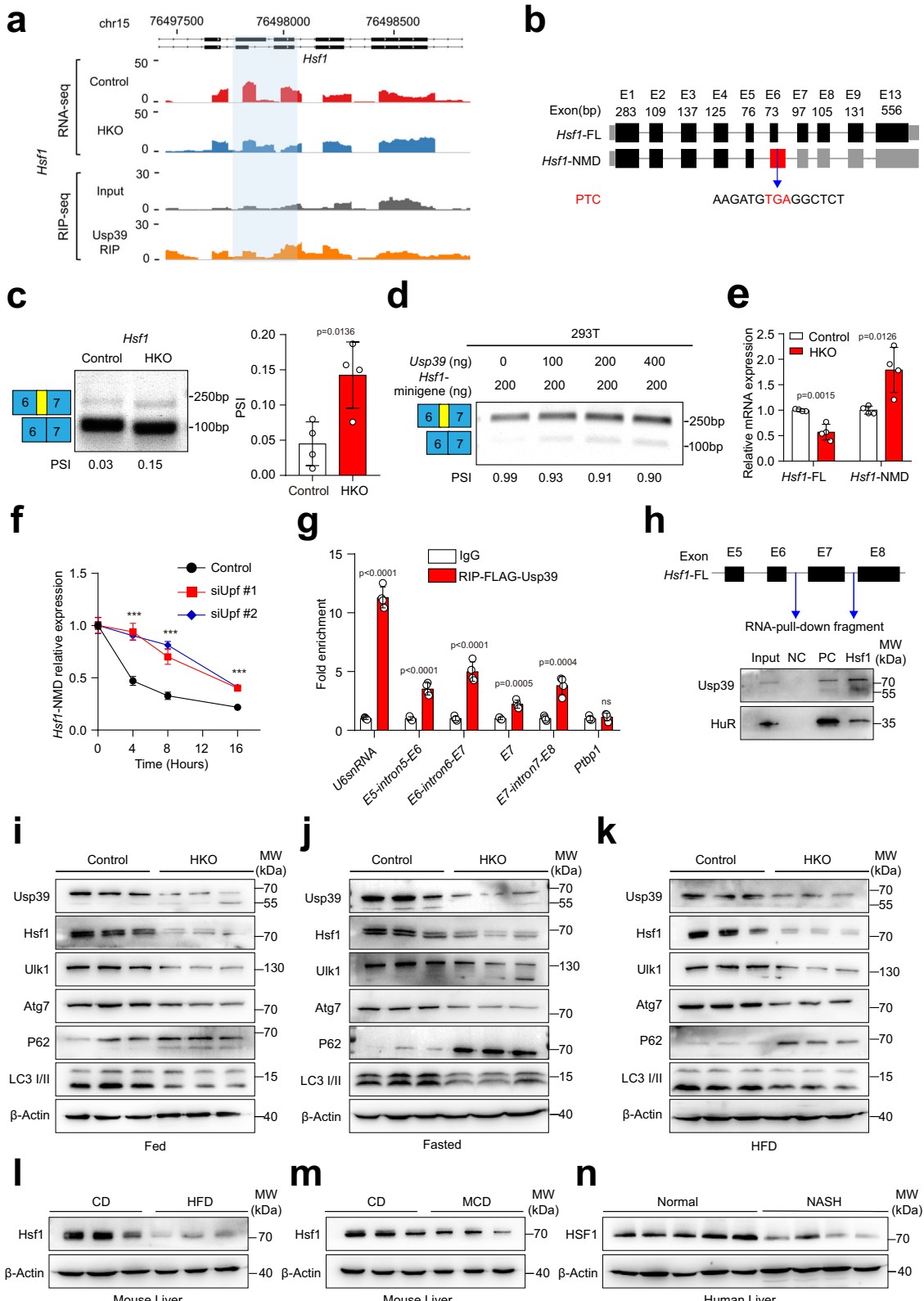

**Transmission electron microscopy**

Mouse livers in 5-week-old *Usp39*-HKO male C57BL/6 mice (*n* = 10) and control (*n* = 11) mice fasted for 16 h were fixed with fixtive for TEM (servicebio) at 4 °C for 30 min. After washing with 0.1 M PB (pH 7.4) three times for 15 min each, the samples were post-fixed with 1% O$_S$O$_4$ in 0.1 M PB (pH 7.4) for 2 h at room temperature. The samples were then dehydrated with ethanol, followed by embedding in acetone.

Ultrathin sections were prepared and stained with uranyl acetate; the samples were imaged with a Hitachi HT7800 TEM.

**RNA-seq, RIP-seq, and bioinformatics**

For RNA-seq, 5-week-old *Usp39*-HKO and control male littermates (*n* = 4) were fasted for 16 h and anesthetized. RNA from liver tissues was extracted with TRIzol and prepared for RNA-seq using an Illumina

**Fig. 6 | *Usp39* deletion leads to mis-splicing and fast degradation of *Hsf1* in hepatocytes. a** AS pattern and Usp39 binding sites on *Hsf1* were visualized with IGV using RNA-seq and RIP-seq data. AS sites and the Usp39 binding region was highlighted. **b** Schematic structure of two *Hsf1* isoforms. *Hsf1*-FL is the canonical isoform, and *Hsf1*-NMD is the altered splicing (A5SS) of exon 6. **c** RT-PCR was performed to validate AS events of *Hsf1* in the livers of 5-week-old male mice. Percent spliced in (PSI) was quantified (*n* = 4). **d** Analysis of the splicing of the *Hsf1* by minigene in 293 T cells. **e** Relative expression of *Hsf1*-FL and *Hsf1*-NMD was analyzed by qPCR in liver tissues (*n* = 4). **f** *Hsf1*-NMD was measured by qPCR in *Upf1* knockdown and control AML12 cells treated with 10 μg/mL actinomycin D. **g** RIP-qPCR was performed to validate the binding of Usp39 on the *Hsf1* transcript in AML12 cells (*n* = 4). U6 snRNA and Ptbp1 were used as the positive and negative controls. **h** RNA pull-down showing the interaction between Usp39 protein and *Hsf1* pre-mRNA. PC: positive control, NC: negative control. Protein was quantified with Input. **i–k** Immunoblotting was performed to measure the levels of indicated genes in *Usp39*-HKO livers compared with controls under fed (**i**), fasted (**j**) and HFD (**k**) conditions (*n* = 3). **l**, **m** Hsf1 level was analyzed by immunoblotting in the livers of MCD-fed (*n* = 6) and HFD-fed (*n* = 7) mice compared to chow-fed mice. **n** HSF1 was measured by immunoblotting in the livers of individuals with NASH (*n* = 4) compared with those without NASH (*n* = 5). Immunoblotting bands intensities were quantified by Image J (Fig. S9a-c, g-i). Images are representative of at least three independent experiments. *P*-value was obtained by unpaired two-sided Student's *t*-test and one-way analysis of variance (ANOVA) followed by Dunnett's multiple comparisons test for (**f**), and the results are presented as the mean ± S.D. **p* < 0.05, ***p* < 0.01, ****p* < 0.001. NS, stands for non-significant. CD, control diet. MW, molecular weight. Source data are provided as a Source data file.

NovaSeq 6000. Sequence data were aligned to the mm10 genome with HISAT2 (version 2.2.0) and sorted with samtools (version 1.9). Mapped reads were visualized with the Integrative Genomics Viewer (IGV). Gene expression was quantified with featurecounts and differential expression was analyzed with DESeq2. The cutoff was set *p* < 0.05 and log2 fold change (FC) > 0.5 or <−0.5. The mapped reads aligned by HISAT2 were further used for AS analysis and the events were identified with rMATS (version 4.1.0)[37]. AS events were classified into skipped exon, retained intron, alternative 5' splice site, alternative 3' splice site, and mutually exclusive exons. The RNA-seq data generated in this study have been deposited in the NCBI GEO database under the accession number GSE213633. Gene expression profile and alternative splicing events of RNA-seq are listed in Supplementary Data 1 and Supplementary Data 2

RIP-seq experiment was conducted using Magna NuCLEAR™ RIP (Cross-Linked) Nuclear RNA-Binding Protein Immunoprecipitation Kit (17-10520) with several modifications to improve the signal resolution. Doxycycline-induced FLAG-Usp39 overexpressing AML12 cell line was constructed. After 72 h of doxycycline induction, the cells were collected and nuclear pellets were isolated with Minute™ Cytoplasmic and Nuclear Extraction Kits for Cells (Invent). Then the nuclear pellets were cross-linked with paraformaldehyde for 5 minutes and then lysed. RNA was fractionated using Bioruptor Pico sonication device (Diagenode) for 10 cycles and incubated with anti-FLAG magnetic beads (MBL, M185-11) at 4 °C overnight. Pull-down RNA fragments after thorough washing were extracted with Spin Column RNA Cleanup & Concentration Kit (Sangon). The sequencing library was constructed with the NEB Next Ultra RNA Library Prep Kit for Illumina, and high-throughput sequencing of the RIP libraries was performed on a NovaSeq 6000 using the PE150 sequencing strategy (Novogene). The adapter sequences and low-quality reads were removed with Trim Galore (version 0.6.1), and the quality of the clean reads was checked with FastQC (version 0.11.9). rRNA sequences were removed with bowtie (version 1.2.3)[38], and the remaining reads were mapped to the mouse genome (mm10) with HISAT2 and visualized in IGV[39]. Usp39 binding peaks were identified with Piranha (-p 0.001 -z 100) and annotated with an R package clusterProfiler. De novo motif analysis was conducted with HOMER. The RIP-seq data generated in this study have been deposited in the NCBI GEO database under the accession number GSE213629. RIP-seq identified Usp39 binding peaks are listed in Supplementary Data 3.

Raw sequencing data of mouse models of NASH and HFD were achieved from obtained from the GEO database (GSE154892 and GSE165855). The splicing factors interaction network was analyzed based on STRING database and visualized with Cytoscape.

### Lipidomics
Lipids were extracted from liver tissues of *Usp39*-HKO and control male mice (*n* = 4) according to Methyl-t-butyl ether (MTBE) method[40]. Briefly, samples were homogenized with 200 μL water and 240 μL methanol, and then 800 μL of MTBE was added and the mixture was ultrasonicated for 20 min at 4 °C followed by incubation for 30 min at room temperature. The solution was centrifuged at 14,000 × *g* for 15 min at 10 °C and the upper organic solvent layer was removed and dried under nitrogen. Reverse phase chromatography was used for LC separation using a CSH C18 column (1.7 μm, 2.1 mm × 100 mm, Waters). The lipid extracts were re-dissolved in 200 μL 90% iso-propanol/acetonitrile and, centrifuged at 14,000 × *g* for 15 min, finally, 3 μL sample was injected. Mass spectra are acquired by Q-Exactive Plus in positive and negative mode. The identification of lipid species based in the MS/MS math data was performed with LipidSearch[41]. Differentially present lipid molecules were selected by Student's *t*-test (*p* < 0.05) and fold change (FC > 1.5 or FC < 0.67), and the lipidomics data were clustered and visualized with R Bioconductor 4.0.2. Lipidomics data are shown in Supplementary Data 4.

### Immunoblotting
Whole cell or tissue protein lysates were prepared from liver tissue or cultured cells using lysis buffer, and an Enhanced BCA Protein Assay Kit (Beyotime Biotechnology) was used for protein concentration measurement. Equal amounts of total protein were loaded on 12% SDS-PAGE gels and transferred to polyvinylidene difluoride (PVDF) membranes. The PVDF membrane was blocked with 5% skim milk for 1 h at room temperature and incubated with diluted primary antibodies overnight at 4 °C. Antibodies that were used included anti-rabbit Usp39 (1:5000 Abcam ab131244), LC3A/B (D3U4C) XP (1:1000 Cell Signaling Technology 12741), SQSTM1/p62 (1:1000 Cell Signaling Technology 5114), Atg7 (1:1000 Cell Signaling Technology 8558), Ulk1 (1:800 Abways CY6902), Hsf1 (1:1000 Proteintech 51034-1-AP), Caspase-3 (1:1000 Cell Signaling Technology 9662), Cleaved Caspase-3 (1:1000 Cell Signaling Technology 9661), Lamin B1 (1:1000 Proteintech 12987-1-AP), Gapdh (1:10,000 Proteintech 10494-1-AP), Usp39 (1:10,000 Abcam ab131332), Usp39 - N - terminal (1:2000 Abcam ab236453), Usp39 (1:2000 Invitrogen A304-816A), anti-mouse Actin (1:5000 Proteintech 66009-1-Ig). The membranes were then conjugated to horseradish peroxidase (HRP)-conjugated secondary antibody (1:10,000, Jackson ImmunoResearch) and the signals were developed with Pierce ECL Substrate (Vazyme, E412-02). Results were analyzed using Image J. The primary antibodies used in this study are shown in Key resources Supplementary Table 3.

### Human liver samples
Human healthy and NASH liver samples were collected from the Department of General Surgery, Shandong University, with participants gave written informed consent, according to CARE guidelines and in compliance with the Declaration of Helsinki principles. Human healthy liver samples from patients who benign liver disease biopsy sample or benign neoplastic pathological changes. The study was approved by the Research Ethics Committee of Shandong University (SDULCLL2019-1-09). Sex was not considered in this study due to the low frequency of subjects. Detailed characteristics of human healthy and NASH patients have been listed in Supplementary Table 4.

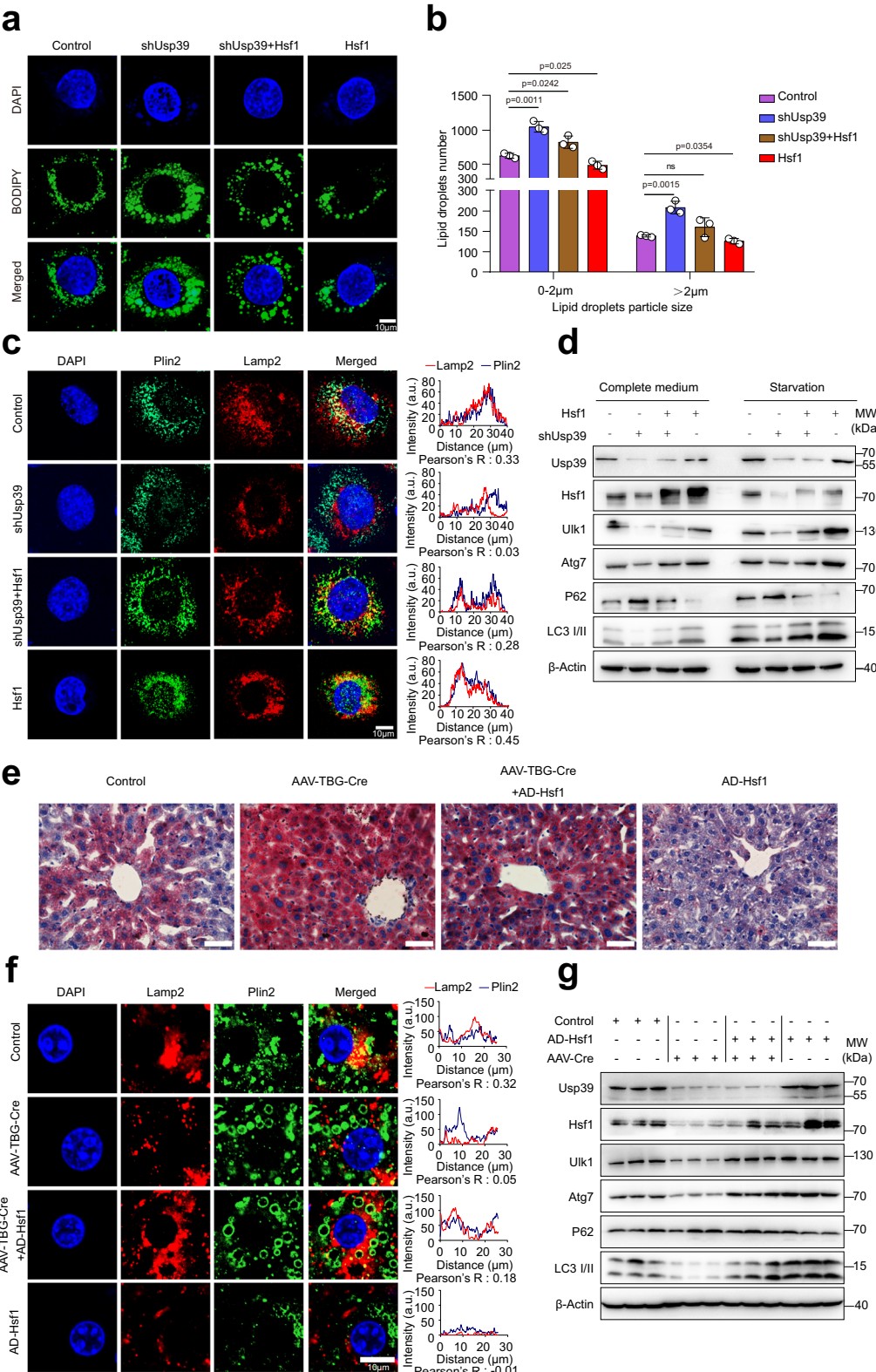

## RNA pull-down assay

*Hsf1* was cloned from mouse liver genomic DNA, and the RNAMAX-T7 in vitro transcription kit (RiboBio, China) was used to transcribe the RNA of the *Hsf1* fragment, followed by biotin labeling using the Pierce™ RNA 3′ End Desthiobiotinylation Kit (Thermo Fisher Scientific). RNA pull-down was performed using the Magnetic RNA-Protein Pull-Down Kit (Thermo Fisher Scientific) according to the manufacturer's instruction and the proteins were detected by immunoblotting.

## Statistical analysis

All data are presented as means ± S.D. Statistical analysis was performed by Student's two-tailed *t*-test or two-way analysis of

**Fig. 7 | Hsf1 promotes autophagy and alleviates hepatic steatosis caused by *Usp39* deletion. a**, **b** AML12 cells transfected with indicated vectors. Cells were stained with BODIPY and imaged using confocal microscopy. Lipid droplet numbers were quantified of 30 BODIPY-stained cells in each group ($n = 3$ independent experiments) by Image J. Scale bar, 10 μm. **c** Co-localization between Plin2 (green) and Lamp2 (red) was analyzed by immunofluorescence staining in AML12 cells transfected with indicated vectors, co-localization signal was quantified by Image J. Scale bars, 10 μm. **d** Immunoblotting analysis of Usp39, Hsf1 and autophagy-related genes in AML12 cell transfected with indicated vectors in complete or serum deprived medium ($n = 3$, 4 per group, from three independent experiments). **e** Oil Red O staining of liver sections of *Usp39*-floxed mice injected with AAV-empty, AAV-TBG-Cre, or Ad-Hsf1. The mice were sacrificed after fasted for 16 h. Scale bars,

50 μm. **f** Immunofluorescence experiments showed co-localization of Plin2 (green) and Lamp2 (red). Scale bar, 10 μm. **g** Immunoblotting analysis of protein levels of indicated genes in the livers from *Usp39*-floxed mice injected with AAV- empty, AAV-TBG-Cre, or Ad-Hsf1. The mice were sacrificed after fasted for 16 h ($n = 3$, 4, 6 per group, from three independent experiments). Immunoblotting band intensities were quantified by Image J, which were shown in Fig. S10d, j. Co-localization was quantified by Image J. Images are representative of at least three independent experiments. *P*-value was obtained by unpaired two-sided Student's *t*-test and the results are presented as the mean ± S.D. NS, stands for non-significant. AD, Adenovirus. AAV, Adeno-associated virus. TBG, thyroxine-binding globulin. MW, molecular weight. Source data are provided as a Source data file.

variance (ANOVA), and all experimental results were repeated at least three times with independent samples. *P* values less than 0.05 were considered statistically significant. The asterisks in the figures indicate statistical significance as *$p < 0.05$, **$p < 0.01$, ***$p < 0.001$. NS, stands for non-significant.

### Reporting summary
Further information on research design is available in the Nature Portfolio Reporting Summary linked to this article.

## Data availability
The RNA-seq, RIP-seq, for this study are available for download from the Gene Expression Omnibus (GEO) repository (GSE213633, GSE213629). Lipidomics data are shown in Supplementary Data 4. Source data are provided with this paper.

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

## Acknowledgements

We thank Translational Medicine Core Facility of Shandong University for consultation and instrument availability that supported this work. We also thank the Model Animal Research Centre of Shandong University for mouse housing and care. This work was supported by the National Natural Science Foundation of China (81972437 to Z.L., 81672578 to Z.L., 32150710523 to C.S.), and Cutting-Edge Development Fund of Advanced Medical Research Institute (Z.L.).

## Author contributions

D.C. performed most experiments, and data analysis, and writing the manuscript. Z.W. performed RNA-seq, RIP-seq, Lipidomics and public database sequence data analysis, involved in the study design, experiments and revised the manuscript. Q.D., J.W., Y.Z., and J.Q. helped with animal and cell experiment. J.S. and X.Z. helped with the human sample collection. L.Z., G.Lu, H.L., R.L., G.Liu is involved technical, material support and guided the experiment. C.S., X.Z., and Z.L. Supervised the project and revised the manuscript. Z.L. was the lead contact for the study. Final approval: All authors.

## Competing interests

The authors declare no competing interests.
