## [Peer Review File · Nature Communications]

Spliceosome component Usp39 contributes to hepatic lipid homeostasis through the regulation of autophagyREVIEWER COMMENTS

Reviewer #1 (Remarks to the Author):

This is a potentially interesting contribution to explore the role of Usp39, a spliceosome component, in the regulation of hepatocyte lipid homeostasis. The authors show that Usp39 level is downregulated in hepatic tissues of non-alcoholic fatty liver disease (NAFLD) and non-alcoholic steatohepatitis (NASH) mice. They make use of a mouse model of NAFLD and conditional knock-out (KO) mouse lines to test their hypothesis. They show that hepatocyte-specific Usp39 deletion leads to increased lipid accumulation, spontaneous steatosis and impaired autophagy. They used combined analysis of RIP-seq and RNA-seq data to show that Usp39 regulates alternative splicing (AS) of several autophagy-related genes. Therefore, reduced expression of these autophagy-related genes may increase steatosis to promote NAFLD. Finally, they show that overexpression of Hsf1, which is downregulated by Usp-39 deficiency, reduces lipid accumulation caused by Usp39 deficiency.

A greater effort is needed to characterize autophagy/lipophagy pathways. I am not sure I understand which pathways we are talking about. There are inconsistencies in the writing of the manuscript. Human data is mandatory to increase the overall significance of this study.

Detailed major concerns:

Related to Methodology:

I do not believe it would be possible to replicate the cell/mice studies based on the information presented.

Related to Figure 1:

Most of these data were collected in mice, not humans. Please make a statement in the abstract for not misleading the reader. To ensure the relevance and reliability of these experimental data, I strongly encourage the authors to study the expression of Usp39 in a cohort of human NAFLD patients.

It is also important to better characterize how these data were collected. What is the difference between HFD and NASH in Fig1A and 1B? What are the experimental mouse models used? Fig1D, I can barely see the accumulation of steatosis in HFD-fed mice, please explain? I am not really convinced by the IHC staining, what cell type expresses USP-39? Where is it located? At what molecular weight Usp39 is expressed?

Related to Figure 2:

As stated by the authors, liver function was restored in Usp39-HKO mice by the age of 8 weeks. How can the authors explain this? Is the expression of Usp39 recovered due to hepatic progenitor cell activation? Please better characterize this possibility. Please show the expression of Usp-39 on WB from E13.5 to 8 weeks. Please better characterize the spontaneous phenotype of these mice by measuring liver injury (Transaminases, LDH, cell death mediators...), inflammation (expression of inflammatory markers...) and fibrosis (SR, collagen expression...).

Related to Figure 3:

H/E staining shown in Fig3B does not reflect the measurement of TG? Is it representative? Please show the expression of Usp39 at this age? Please explain why the mice were fasted 16 hours before performing the lipidomic experiments? Your hypothesis is that autophagy is inhibited in KO mice, but by fasting the mice, you trigger autophagy (in CTL mice), thus increasing the difference between CTL and KO mice. Were the experiments performed without fasting?

Related to Figure 4:

Data are poorly described into the manuscript (Global statement). Lipophagy is the most effective way to mobilize fatty acids. To better understand the impact of the modulation of autophagy by Usp39 on cell metabolism, please measure the oxygen consumption (OCR) by Seahorse. Better use oleic acid-treated primary hepatocytes rather than AML-12 cells.

Related to Figures 5-7:

Please explain why an inducible model is used for the rescue experiments? The phenotype of these mice is poorly described. The oil-red O staining shows nothing obvious. Please further characterize this mouse model. Please provide a semi-quantitative analysis of all immunofluorescence images.

Minor concerns:

Introduction

Lines 42 to 53:

I am not sure I understood all the information in this paragraph, please reformulate it.

Reviewer #2 (Remarks to the Author):

This manuscript, by using appropriated methodology, has unveiled a previously unknown role of the spliceosome component Usp39 in hepatocytes by modulating autophagy. Importantly, its expression is downregulated in hepatic tissues of NAFLD and NASH subjects. This led to the generation of hepatocyte-specific mice that presented enhanced lipid accumulation and decreased autophagy-related gene expression. Mechanistically, it was found that Usp39 regulates alternative splicing of relevant autophagy-related genes and its deletion resulted in alternative 5' splice site selection of exon 6 in Hsf1 and, consequently, reduced expression. Overexpression of Hsf1 attenuated lipid accumulation caused by Usp39 deficiency pointing to the unique role of Usp39-mediated alternative for sustaining autophagy/lipophagy and lipid homeostasis in the liver.

In general, the topic of the study is novel, and data are aligned with the conclusions. However, there are several flaws that must be considered in order to strength the conclusion of this study, particularly related to the characterization of lipophagy.

Specific points:

- In all the panels of each figure the number of independent experiments/mice used must be indicated.
- The authors must justify the different pattern of bands in the USO39 western blots in the manuscript. In some blots there are 2 bands. Also, quantification and statistical analysis in graphs with individual points is needed.
- Figure 1. Do the authors find differences in food intake?
- Figure 2H Quantification of PCNA positive cells must be included.
- The authors claim that liver function was restored in Usp39 hepatocyte KO mice at 8 weeks. This is only assessed by HE staining. A more complete set of parameters of liver function must be included (e.g. liver enzymes). Also, markers of immune cell infiltration must be evaluated.
- Figure S2F. Liver TG in female mice is missing.
- Figure S3E. Energy Expenditure, RER and locomotor activity are missing.
- Figure 4G only shows a representative image. Quantification is needed.
- Figure 7, as stated above quantification is needed. In the work of Mc Niven's group (<https://doi.org/10.1083/jcb.201803153>) they demonstrated Lamp2 associated with lipid droplets by several techniques.

Reviewer #3 (Remarks to the Author):

This article presents evidence that the deubiquitinase Usp39 regulates NAFLD and NASH phenotypes including steatosis by regulating alternative splicing of Hsf1. Usp39 has been ascribed roles outside the spliceosome in regulating Nfkb, cell proliferation and other phenotypes. Parts of the phenotypes should be expected to be independent of alternative splicing, and alternative splicing of Hsf1 in particular. Hsf1 effects in addition should be very broad beyond autophagy. The authors show reproducible effects of Usp39 on levels of multiple autophagy proteins and important effects on liver phenotypes that resemble those reported for autophagy-deficient mice. Overall the authors have presented an extensive amount of evidence linking Usp39 to these liver phenotypes and regulation of autophagy. There are many aspects of the article that rely on representative microscopy images or other representative data that require more in-depth quantification. In general, some of the links in RNAseq data to autophagy are not striking and the phenotypes on lipophagy as the key mechanism are inconclusive. The authors should moderate their conclusions regarding these data and improve quantitation of them.

1. Fig.1 Is the regulation of Usp39 in microarray data and in Western blot data in mice and human samples statistically significant?
2. From the microarray data, Usp39 does not seem the most consistently or strongly regulated mRNA, even within the spliceosome complex. Why did the authors focus on Usp39 based on this data?
3. There are two bands for Usp39 in Western blots in Fig.1 | MCD and HFD conditions. Is this Usp39 as well? Are these Usp39 splice variants or otherwise?
4. Fig.2H. Differences in Ki67 expression are not apparent.
5. Are the authors certain that all the differences observed are due to expression of the transgene in hepatocytes? Is the albumin promoter really so liver-specific or could some of the differences be due to expression for example in subsets of immune cells?
6. Fig.3 – RNAseq data must be presented in alternative ways including with a heatmap of identified mRNAs showing differential expression between wild-type and HKO. GO-term or Ingenuity pathway mapping of hepatocyte-specific genes could easily lead to these types of GO-terms being pronounced.

7. Fig.4 – Increased levels of p62 protein and decreased LC3-II could indicate impaired autophagy. This result could have alternative explanations, like increased mRNA expression or translation of p62 and decreased LC3A,B,C mRNA. To be more convincing that there are changes in autophagosomes by electron microscopy this should be quantified, as should changes in LC3 punctae by fluorescent microscopy. LAMP2 decreases are not necessarily indicative of effects on autophagy, and are not convincing in a single panel of microscopy. The authors should also quantify p62 and LC3B mRNA by RT-qPCR.

8. Why did the authors focus on the levels of these mRNAs (Atg3, Atg7, Ulk1) among tens of other autophagy-related mRNAs? Are these Hsf1 targets?

10. Fig.5. The mRNAs focused on from the RNAseq and RIP analysis are fairly indirect regulators of autophagy. If this is the full extent of autophagy-regulators identified I am somewhat skeptical of the overall dysregulation/impact of autophagy. Why or how these would affect the mRNA levels of Atg3, Atg7 etc. is not clear. Usp39 binding by RIP seems to parallel total RNAseq quite closely and bind a large proportion of mRNAs. The specificity of this RIP analysis is doubtful and is not made better by weak results showing specificity for Exon 6-7 region of Hsf1 in Fig.6. Splicing differences identified by RT-PCR should be quantified over multiple replicates.

11. What was the average fragment length in RIP analysis? This may impact the relevance of the motifs identified. Are these motifs enriched in autophagy-related mRNAs? Near the alternative splice sites? Without further analysis this is out questionable value.

12. Differences in levels of Hsf1 protein are notable between wild-type and HKO conditions across diets. This regulation could occur through multiple pathways. The authors should confirm differences in splicing in these different conditions quantified across multiple mice as in Fig.6C or E.

13. Fig.7 contains what would be critical experiments to demonstrate that Usp39 mediates effects by regulating Hsf1. Differences in lipid accumulation and lipophagy must be quantified in a robust manner – representative images are not adequate. Co-localization of PLIN2 with LC3 or LAMP2 could be due to decreased degradation of lipid droplets after targeting to autophagosomes in addition to increased rate of lipophagy. There could also be a lipophagy-independent mechanism that is still autophagy or lysosome-dependent. Conclusions should be tempered.

14. How conserved are the sequences in the regions spliced in the mouse liver and mouse hepatocyte cells with human sequences? The authors should confirm that Usp39 regulates this splicing event in a human hepatocyte line or primary hepatocyte (e.g. HepG2 or Huh7?) to support the splicing mechanism being the relevant one in humans to support downregulation of Hsf1 in NASH patients.

Response to reviewers' comments

We thank the reviewers for their constructive and thoughtful comments, which have helped us to improve the manuscript substantially. According to the reviewers' comments, we have made extensive modifications to our manuscript and supplemented extra data to further substantiate our findings. Changes to the manuscript are shown in red. The following is a point-by-point response to the reviewers' comments.

Response to Reviewer #1 (Remarks to the Author)

-----Reviewer comments:

This is a potentially interesting contribution to explore the role of Usp39, a spliceosome component, in the regulation of hepatocyte lipid homeostasis. The authors show that Usp39 level is downregulated in hepatic tissues of non-alcoholic fatty liver disease (NAFLD) and non-alcoholic steatohepatitis (NASH) mice. They make use of a mouse model of NAFLD and conditional knock-out (KO) mouse lines to test their hypothesis. They show that hepatocyte-specific Usp39 deletion leads to increased lipid accumulation, spontaneous steatosis and impaired autophagy. They used combined analysis of RIP-seq and RNA-seq data to show that Usp39 regulates alternative splicing (AS) of several autophagy-related genes. Therefore, reduced expression of these autophagy-related genes may increase steatosis to promote NAFLD. Finally, they show that overexpression of Hsfl, which is downregulated by Usp-39 deficiency, reduces lipid accumulation caused by Usp39 deficiency.

A greater effort is needed to characterize autophagy/lipophagy pathways. I am not sure I understand which pathways we are talking about. There are inconsistencies in the writing of the manuscript. Human data is mandatory to increase the overall significance of this study.

Response: We thank the reviewer for the constructive comments that helped us to improve the manuscript. To better characterize autophagy/lipophagy pathways, we performed additional experiments including EGFP-LC3 and BODIPY co-localization

staining and found that co-localization signals were significantly reduced upon Usp39 knockdown in primary hepatocytes and AML12 cells (Revised Fig. 4i, j and Fig.s4 i, j). These data, together with our previous results indicate that impairment of macroautophagy is responsible for the lipid accumulation caused by Usp39 deficiency. As lipophagy may occur via macrolipophagy, microlipophagy, or chaperone-mediated autophagy (CMA) (Schulze R, PNAS. 2020; Zhang Q, Cell Rep. 2023) ^{1,2}, our study does not rule out roles of possible defects in microlipophagy or CMA in lipid accumulation caused by Usp39 deficiency. To accurately describe our findings, we have replaced “lipophagy” with “autophagy” in the revised manuscript.

As suggested by the reviewer, we have performed additional expression analysis of Usp39 in a cohort of human NAFLD patients (GSE193084) (Revised Fig. 1d-f, as shown below). The datasets were from Fujiwara N, Sci Transl Med. 2022 ³.

Our responses to specific concerns are given below.

Specific points:

1.Related to Methodology: I do not believe it would be possible to replicate the

cell/mice studies based on the information presented.

Response: We thank the reviewer for pointing this out. We have provided more detailed information on this section.

2. Related to Figure 1:

1) Most of these data were collected in mice, not humans. Please make a statement in the abstract for not misleading the reader.

Response: We are grateful to your comments and suggestions. We have stated that most of findings were collected in mice as suggested.

2) To ensure the relevance and reliability of these experimental data, I strongly encourage the authors to study the expression of Usp39 in a cohort of human NAFLD patients.

Response: This suggestion is highly appreciated. We have performed additional expression analysis of Usp39 in a cohort (GSE193084) of human NAFLD patients (Fujiwara N, Sci Transl Med. 2022) ³. The results showed that Usp39 expression was negatively correlated with NAFLD activity score (Revised Fig.1d-e). Consistently, Usp39 level was lower in high fibrosis stage (2-4) than in low fibrosis stage (0-1) (Revised Fig. 1f and Table.S8).

3) It is also important to better characterize how these data were collected. What is the difference between HFD and NASH in Fig1A and 1B? What are the experimental mouse models used?

Response: The transcriptomic data of Fig1a was generated from GEO database GSE165855 (Tang, X, J Biol Chem. 2021) ⁴. It is a liver transcriptomic dataset from chow-fed and HFD-fed (60% kcal from fat) mice for 4 months (n=6). The transcriptomic data of Fig1b was from GEO database GSE154892 (Flores-Costa R, PNAS. 2020) ⁵, which is a liver transcriptomic data from chow-fed and CDAHFD (choline-deficient l-amino acid-defined high-fat diet)-induced NASH model (n=8). We have modified the manuscript and added references accordingly.

4) Fig1D, I can barely see the accumulation of steatosis in HFD-fed mice, please explain?

Response: We thank the reviewer for the insight. We apologize for not providing a better representative image and have included new representative images in the revised manuscript (Revised Fig. 1g).

5) I am not really convinced by the IHC staining, what cell type expresses USP-39? Where is it located? At what molecular weight Usp39 is expressed?

Response: We apologize for the poor IHC staining. We have optimized IHC assay and replaced the IHC staining result with images of higher quality (Revised Fig. 1g), which showed that Usp39 is predominantly localized in the nucleus in the control group, but is remarkably reduced in intensity in the HFD or MCD mice. To confirm the nuclear location of Usp39, nuclear and cytosolic fractions from primary hepatocytes were subjected to western blot analysis and the result showed that Usp39 was predominantly localized in the nucleus (Revised Fig.S1e). Furthermore, we performed immunofluorescence assay in primary hepatocytes and found that Usp39 was colocalized with SC35, which is a marker of the nuclear speckle (Revised Fig. S1f). The full-length Usp39 consists of 564 amino acids and has a calculated molecular mass of 65 kD. Interestingly, a 55 kD band was also detected both in liver tissues and cell lines by western blot for each of three different antibodies used (Figure 1 for reviewer). The function of this 55 kD band is currently beyond our knowledge, but we do not believe that the presence of this extra band would affect the conclusion drawn in this study.

3. Related to Figure 2:

As stated by the authors, liver function was restored in Usp39-HKO mice by the age of 8 weeks. How can the authors explain this? Is the expression of Usp39 recovered due to hepatic progenitor cell activation? Please better characterize this possibility. Please show the expression of Usp-39 on WB from E13.5 to 8 weeks. Please better characterize the spontaneous phenotype of these mice by measuring liver injury (Transaminases, LDH, cell death mediators...), inflammation (expression of inflammatory markers...) and fibrosis (SR, collagen expression...).

1) **Response:** We thank the reviewer for raising this concern. Per your comments, we first measured protein level in control and *Usp39*-HKO mice at different ages (5 week, 10week and 12 month) and found the expression of *Usp39* was not recovered (Figure 2 for reviewer). Second, immunofluorescence staining of Sox9 in liver tissues showed that no Sox9 positive cells were found in the liver of *Usp39*-HKO mice, suggesting that no hepatic progenitor cells are activated (Fig. S2j, k). We further measured *Usp39* expression by qPCR and immunoblotting and found the expression level of *Usp39* was gradually reduced in the mice liver during development (Revised Fig. S-1g, h).

Following your suggestion, we have also measured several markers (ALT, AST, LDH, albumin levels and immune cell infiltration) of liver function. The results showed that at the age of 5 weeks, *Usp39* deletion induced spontaneous liver injury, as revealed by increased level of ALT, AST and LDH (Revised Fig. 2n) and fibrosis, as assessed by Sirius red staining and qPCR (Fig. 2o-p). Interestingly, liver injury was significantly alleviated in 10-week-old *Usp39*-HKO mice compared to 5-week-old *Usp39*-HKO mice (Revised Fig.S2g-p compared to Revised Fig. 2i-p).

These data, together with our previous results demonstrate that *Usp39* deletion induced spontaneous liver injury and fibrosis at young age, suggesting the essential role of *Usp39* in early postnatal liver development in mice. We have modified the description in the revised manuscript accordingly.

4. Related to Figure 3:

1) H/E staining shown in Fig3B does not reflect the measurement of TG? Is it representative?

Response: We thank the reviewer for pointing this out. We have performed new experiment to measure TG level in control and HKO mouse liver samples and obtained similar result. We have now provided new representative images to show the comparable levels in the different assays (Revised Fig. 3b-c).

2) Please show the expression of Usp39 at this age?

Response: As suggested by the reviewer, we performed immunoblotting in the liver tissues of 12-month-old control and *Usp39*-HKO mice and found Usp39 to be significantly reduced upon Usp39 knocking out (Revised Fig. S3d).

3) Please explain why the mice were fasted 16 hours before performing the lipidomic experiments? Your hypothesis is that autophagy is inhibited in KO mice, but by fasting the mice, you trigger autophagy (in CTL mice), thus increasing the difference between CTL and KO mice. Were the experiments performed without fasting?

Response: We thank the reviewer for the insightful comment. We performed lipidomic experiments when we observed increased lipid accumulation in *Usp39*-HKO compared to control mice. Autophagy defect in response to Usp39 knockout was found afterwards. Fasting is indeed a commonly performed procedure to reduce variability in many experimental settings.

According to the reviewer's suggestion, we have included data in control and *Usp39*-HKO mice without fasting. The data still indicate an increased lipid accumulation in *Usp39*-HKO livers compared with controls as evidenced by (Revised Fig. S3e-n). In parallel, hepatic TG level was higher in *Usp39*-HKO mice than in control mice. As correctly pointed by the reviewer, fasting increased the difference between control and *Usp39*-HKO mice.

5. Related to Figure 4:

Data are poorly described into the manuscript (Global statement). Lipophagy is the most effective way to mobilize fatty acids. To better understand the impact of the modulation of autophagy by Usp39 on cell metabolism, please measure the oxygen consumption (OCR) by Seahorse. Better use oleic acid-treated primary hepatocytes rather than AML-12 cells.

Response: Thank you for pointing this problem. We have restructured this section in the revised manuscript. Following your suggestion, we have measured the oxygen consumption rates (OCR) by Seahorse in control and oleic acid-treated primary hepatocytes and AML12 cells with or without Usp39 knockdown. We found that OCR was significantly decreased in Usp39 knockdown cells (Revised Fig. S4l, m), implying an important role of Usp39 in maintaining metabolic homeostasis.

6. Related to Figures 5-7:

Please explain why an inducible model is used for the rescue experiments? The phenotype of these mice is poorly described. The oil-red O staining shows nothing obvious. Please further characterize this mouse model. Please provide a semi-quantitative analysis of all immunofluorescence images.

Response: Thank you for raising this concern. We chose inducible model (AAV-TBG-Cre) in rescue experiments for two reasons. First, Alb-Cre resulted in a long-term deletion of Usp39 starting at as early as embryonic stage in *Usp39*-HKO mice. We speculated that a long-term knockout mouse model might involve more difficulties for rescue experiments. Second, AAV-TBG-Cre is widely used in order to generate hepatocyte specific deletion of genes in adult mice (Zheng Sun, Nature Medicine. 2012) ⁶. The adeno-associated virus encoding TBG-Cre is commercially available.

As suggested by the reviewer, we have provided semi-quantitative analysis of all immunofluorescence images and restructured Figure 7. Furthermore, we have added more data that characterize the phenotype of AAV-TBG-Cre induced hepatocyte specific deletion of Usp39. Usp39-floxed mice injected with AAV- empty and AAV-TBG-Cre followed by a normal chow diet for 4 weeks. We did not observe any differences in body weights, liver weight and liver/body weight ratio (Revised Fig. S7e). Consistent with Oil

Red O staining, we observed pale-colored livers and the hepatic TG level was significantly increased in mice lacking hepatic Usp39 compared to control mice (Revised Fig. S7f, g).

7. Minor concerns:

Introduction

Lines 42 to 53:

I am not sure I understood all the information in this paragraph, please reformulate it.

Response: According to the reviewers' comment, we have restructured this paragraph for better understanding.

Response to Reviewer #2 (Remarks to the Author)

-----Reviewer comments:

This manuscript, by using appropriated methodology, has unveiled a previously unknown role of the spliceosome component Usp39 in hepatocytes by modulating autophagy. Importantly, its expression is downregulated in hepatic tissues of NAFLD and NASH subjects. This led to the generation of hepatocyte-specific mice that presented enhanced lipid accumulation and decreased autophagy-related gene expression. Mechanistically, it was found that Usp39 regulates alternative splicing of relevant autophagy-related genes and its deletion resulted in alternative 5' splice site selection of exon 6 in Hsf1 and, consequently, reduced expression. Overexpression of Hsf1 attenuated lipid accumulation caused by Usp39 deficiency pointing to the unique role of Usp39-mediated alternative for

sustaining autophagy/lipophagy and lipid homeostasis in the liver.

In general, the topic of the study is novel, and data are aligned with the conclusions. However, there are several flaws that must be considered in order to strength the conclusion of this study, particularly related to the characterization of lipophagy.

Response: We thank the reviewer for the appreciation of our work and the valuable comments that help us improve the manuscript. We have performed additional experiments to strengthen the relationship between Usp39 and autophagy mediated lipid droplet catabolism.

Our responses to specific concerns are given below.

Specific points:

1. In all the panels of each figure the number of independent experiments/mice used must be indicated.

Response: According to the reviewer's suggestion, we have now provided the number of independent experiments/mice in all the panels of each figure.

2. The authors must justify the different pattern of bands in the USP39 western blots in the manuscript. In some blots there are 2 bands. Also, quantification and statistical analysis in graphs with individual points is needed.

Response: Thank you for this insightful comment, which was also raised by other two reviewers. Accordingly, we conducted additional western blot assay by using three antibodies from different companies. Intriguingly, the 55kD band was always detected in addition to the full-length Usp39 band (consists of 564 amino acids and has a calculated molecular mass of 65 kD). As in our response to Reviewer #1, its function remains to be determined.

Following your suggestion, we have conducted quantification and statistical analysis in all graphs with individual points.

3. Figure 1. Do the authors find differences in food intake?

Response: The reviewer probably meant Figure 2 instead of Figure 1. We have measured food intake in 5 and 10-week-old *Usp39*-HKO and control mice. 5-week-old *Usp39*-HKO

mice consumed less food than control mice. In contrast, no difference in food intake was observed between 10-week-old *Usp39*-HKO and control group mice (Fig.S4d-e).

4. Figure 2H Quantification of PCNA positive cells must be included.

Response: We have included quantification of PcnA and Ki67 as suggested in the revised manuscript (Fig.2h-j).

5. The authors claim that liver function was restored in *Usp39* hepatocyte KO mice at 8 weeks. This is only assessed by HE staining. A more complete set of parameters of liver function must be included (e.g. liver enzymes). Also, markers of immune cell infiltration must be evaluated.

Response: We thank the reviewer for raising this concern. Following your suggestion, we have measured additional parameters (ALT, AST, LDH, albumin levels and immune cell infiltration) of liver function. The results showed that the livers in adult *Usp39*-HKO mice were healthier than those in young *Usp39*-HKO mice (Revised Fig.2n-p and Fig.S2g-p). Based on these data, we propose that *Usp39* plays important role in early postnatal liver development in mice. We have modified the description in the revised manuscript.

6. -Figure S2F. Liver TG in female mice is missing.

Response: Thanks for the reminder. We have provided liver TG in 10-week-old control and *Usp39*-HKO female and male mice (Revised Fig. S3k-n).

7. -Figure S3E. Energy Expenditure, RER and locomotor activity are missing.

Response: According to the reviewer’s suggestion, we have now provided data on Energy, Expenditure, RER and locomotor activity in 5- and 10-week-old control and HKO mice. No significant difference in Energy Expenditure, RER were found between control and HKO mice at the age of 5-week-old (Revised Fig. S4d, e). However, the locomotor activity of HKO mice was significantly lower than that of control mice at 10-week-old. We have included the new data and revised the manuscript accordingly.

8. -Figure 4G only shows a representative image. Quantification is needed.

Response: We have included quantification of original Figure 4G (Revised Fig.4h) as suggested in the revised manuscript.

9. Figure 7, as stated above quantification is needed. In the work of Mc Niven's group (<https://doi.org/10.1083/jcb.201803153>) they demonstrated Lamp2 associated with lipid droplets by several techniques.

Response: Thanks for suggesting the new techniques. We have included quantification for each panel of Figure 7. In addition to Plin2 and Lamp2 co-localization, we also performed EGFP-LC3 and BODIPY co-localization staining and found co-localization signal was significantly reduced upon Usp39 knockdown in primary hepatocytes and AML12 cells (Revised Fig.4i, j and Fig.S4i, j). These results indicate that impaired autophagy is the main cause of hepatic lipid accumulation upon *Usp39* depletion.

Response to Reviewer #3

-----Reviewer comments:

This article presents evidence that the deubiquitinase Usp39 regulates NAFLD and NASH phenotypes including steatosis by regulating alternative splicing of Hsf1. Usp39 has been ascribed roles outside the spliceosome in regulating NfκB, cell proliferation and other phenotypes. Parts of the phenotypes should be expected to be independent of alternative splicing, and alternative splicing of Hsf1 in particular. Hsf1 effects in addition should be very broad beyond autophagy. The authors show reproducible effects of Usp39 on levels of multiple autophagy proteins and important effects on liver phenotypes that resemble those reported for autophagy-deficient mice. Overall, the authors have presented an extensive amount of evidence linking Usp39 to these liver phenotypes and regulation of autophagy. There are many aspects of the article that rely on representative microscopy images or other representative data that require more in-depth quantification. In general, some of the links in RNAseq data to autophagy are not striking and the phenotypes on lipophagy as the key mechanism are inconclusive. The authors should moderate their conclusions regarding these data and improve quantitation of them.

Response: We thank the reviewer for kindly reviewing our manuscript and making the many constructive comments. According to your suggestions, we have included quantification of microscopy images or other representative data. We have also optimized bioinformatics analysis of RNA-seq data and performed new experiments related to autophagy to substantiate our conclusions. In addition, we revised the text to avoid overclaim.

Our point-to-point responses are given below.

1. Fig.1 Is the regulation of Usp39 in microarray data and in Western blot data in mice and human samples statistically significant?

Response: We have conducted statistical analysis of transcriptomic data and quantified western blot data, the downregulation of Usp39 in disease groups was statistically significant both at RNA and protein levels (Revised Fig.1h, i, j and Fig.S1a, b).

2. From the microarray data, Usp39 does not seem the most consistently or strongly regulated mRNA, even within the spliceosome complex. Why did the authors focus on Usp39 based on this data?

Response: We thank the reviewer for pointing this out. Among 16 genes encoding U4/U6. U5 tri-snRNP components, five genes (Usp39, Txnl4a, Prpf4, Sart1, Ddx23) were significantly decreased in the liver of HFD-fed mice compared to chow-fed mice, four genes (Usp39, Prpf6, Prpf8, Ddx23) were significantly decreased in the liver of CDAHFD-fed mice compared to chow-fed mice. We found that Usp39 and Ddx23 were significantly downregulated in RNA-seq data of both HFD and NASH mice. Interestingly, Usp39 expression was also negative correlated with NAFLD activity score in a cohort (GSE193084) of human NAFLD patients (Revised Fig.1d-f and Table.S8). We therefore focused on Usp39 for further investigation. We have remade Figure 1 and added this information to the revised manuscript.

3. There are two bands for Usp39 in Western blots in Fig.1 I MCD and HFD conditions. Is this Usp39 as well? Are these Usp39 splice variants or otherwise?

Response: Thank you for this insightful comment, which was also raised by other two

reviewers. Accordingly, we performed additional Western blot analysis by using three Usp39 antibodies from different companies and found that the 55kD band was always present in addition to full-length Usp39 (consists of 564 amino acids and has a calculated molecular mass of 65 kD) (Figure 1 for reviewer). Next, we analyzed liver transcriptomic data from chow-fed and HFD-fed mice from GEO database under accession number GSE165855. Surprisingly, a short isoform with alternative promoter was identified in the liver of HFD-fed mice (Figure 3 for reviewer). Just as the reviewer speculated, it might represent a new splice variant. While its function is worthy of further exploration, it seems to be beyond the scope of this manuscript. Such investigations will be part of a future manuscript.

4. Fig.2H. Differences in Ki67 expression are not apparent.

Response: We agree with the reviewer on this point. PcnA was more sensitive than Ki67 Based on our experience. We have quantified both Ki67 and PcnA in the revised manuscript (Revised Fig.2h-j).

5. Are the authors certain that all the differences observed are due to expression of the transgene in hepatocytes? Is the albumin promoter really so liver-specific or could some of the differences be due to expression for example in subsets of immune cells?

Response: We thank the reviewer for raising this concern. Albumin-Cre mice have been

widely used to generate hepatocyte-specific gene deletion (He A, Mol Cell, 2020)⁷. To confirm the phenotype was due to hepatocyte-specific deletion of Usp39, AAV-TBG-Cre containing the hepatocyte-specific thyroxin-binding globulin (TBG) promoter was used in this study. Usp39 deletion induced by AAV-TBG-Cre also caused lipid accumulation in the liver of adult mice (Revised Fig.7e and Fig.S7e-g).

6. Fig.3 – RNAseq data must be presented in alternative ways including with a heatmap of identified mRNAs showing differential expression between wild-type and HKO. GO-term or Ingenuity pathway mapping of hepatocyte-specific genes could easily lead to these types of GO-terms being pronounced.

Response: Thank you for the suggestion. We have added the heatmaps and Circos plot to show the differential expression of genes related to liver steatosis and autophagy pathways (Revised Fig. 3m, n and Fig.5b).

7. Fig.4 – Increased levels of p62 protein and decreased LC3-II could indicate impaired autophagy. This result could have alternative explanations, like increased mRNA expression or translation of p62 and decreased LC3A,B,C mRNA. To be more convincing that there are changes in autophagosomes by electron microscopy this should be quantified, as should changes in LC3 punctae by fluorescent microscopy. LAMP2 decreases are not necessarily indicative of effects on autophagy, and are not convincing in a single panel of microscopy. The authors should also quantify p62 and LC3B mRNA by RT-qPCR.

Response: Thank you for the critical comment. We have quantified the number of autophagosomes of electron microscopy data as suggested (Revised Fig. 4g). Further,

EGFP-LC3 was visualized using confocal microscopy and EGFP-LC3 in punctae was quantified. The result showed that *Usp39* deficiency significantly decreased the number of EGFP-LC3 punctae (Revised Fig. 4i, j and Fig.S4i, j).

Immunoblot analysis of LC3 and p62/SQSTM1 has been widely used to monitor autophagic flux (Jiang p, Methods. 2015) ⁸. We have also measured p62 and LC3B mRNA level by RT-qPCR and but observed no significant difference between control and *Usp39*-HKO livers (Figure 4 for reviewer).

8. Why did the authors focus on the levels of these mRNAs (*Atg3*, *Atg7*, *Ulk1*) among tens of other autophagy-related mRNAs? Are these *Hsf1* targets?

Response: Thank you for your comment. *Atg3*, *Atg7*, *Ulk1* are commonly used markers of autophagy (Byun S, Nat Commun. 2020) ⁹, which were significantly down-regulated in our RNA-seq (Revised Fig.5b). We measured their mRNA levels to confirm the role of *Usp39* in the regulation of autophagy. Among these three autophagy-related genes, *Atg7* has been reported to be as one of *Hsf1* targets (Desai S, J Biol Chem. 2013) ¹⁰. In addition, there are other autophagy-related genes, *Atg4b* (Zhang N, Cancer Lett. 2017) ¹¹, *Atg5* and *Atg12* (Cui T, J Invest Dermatol. 2022), that have been reported to be *Hsf1* targets.

9. Fig.5. The mRNAs focused on from the RNAseq and RIP analysis are fairly indirect regulators of autophagy. If this is the full extent of autophagy-regulators identified I am somewhat skeptical of the overall dysregulation/impact of autophagy. Why or how these would affect the mRNA levels of *Atg3*, *Atg7* etc. is not clear. *Usp39* binding by RIP

seems to parallel total RNAseq quite closely and bind a large proportion of mRNAs. The specificity of this RIP analysis is doubtful and is not made better by weak results showing specificity for Exon 6-7 region of Hsf1 in Fig.6. Splicing differences identified by RT-PCR should be quantified over multiple replicates.

Response: Thank you for raising this concern. We integrated Usp39-bound genes and AS- related genes using RIP-seq and RNA-seq data to identify critical target genes of Usp39. Among the candidate genes, Hsf1-was verified as direct targets of Usp39. We propose Hsf1 is one of the key upstream factors that drive transcriptional program of autophagy-related genes including Atg7, Atg4b, Atg5 and Atg12. Overexpression of Hsf1 is sufficient to induce autophagy in *c. elegans* (Kumsta C, Nat Commun. 2017) ¹². Therefore, a large number of dysregulated genes are probably not direct targets of Usp39.

We agree with the reviewer that RIP analysis has limitation in determining the specific sites of interaction of an RNA binding protein with target RNAs. We added heatmaps showing the normalized signal intensity of Input and RIP sample around Usp39 binding sites (Revised Fig.S5a) and splicing sites (Revised Fig. S5b) to confirm the specificity of the RIP-seq experiment. We also found the RIP-seq signal intensity enrichment is more significant around the splicing site of autophagy-related genes (Revised Fig. S5c). To provide further evidence for the direct binding of Usp39 to Hsf1, RNA pull-down and RIP-qPCR assays were performed (Revised Fig. 6g and h). We have also quantified the splicing differences identified by RT-PCR as suggested (Revised Fig. 6c).

10. What was the average fragment length in RIP analysis? This may impact the relevance of the motifs identified. Are these motifs enriched in autophagy-related mRNAs? Near the alternative splice sites? Without further analysis this is out questionable value.

Response: We thank the reviewer’s insight into this matter. The average fragment length is 200-500bp according to the gel image during RIP-seq experiment (Figure 5 for reviewer). Accordingly, we analyzed the two potential binding motifs of Usp39 based on RIP-seq data, and found that the motif1 was enriched in autophagy related genes. Further, the Usp39 binding sites highly enriched in exon-intron regions near splicing sites (Revised Fig.S5d-f). We have included the new data and revised the manuscript accordingly. -

11. Differences in levels of Hsf1 protein are notable between wild-type and HKO conditions across diets. This regulation could occur through multiple pathways. The authors should confirm differences in splicing in these different conditions quantified

across multiple mice as in Fig.6C or E.

Response: Following the advice of the reviewer, we have included the new data with quantification across multiple mice to confirm differences in Hsf1 splicing in HFD-fed and MCD-fed control and *Usp39*-HKO mice. (Revised Fig.S6b-e).

12. Fig.7 contains what would be critical experiments to demonstrate that *Usp39* mediates effects by regulating *Hsf1*. Differences in lipid accumulation and lipophagy must be quantified in a robust manner – representative images are not adequate. Co-localization of PLIN2 with LC3 or LAMP2 could be due to decreased degradation of lipid droplets after targeting to autophagosomes in addition to increased rate of lipophagy. There could also be a lipophagy-independent mechanism that is still autophagy or lysosome-dependent. Conclusions should be tempered.

Response: According to the reviewer’s suggestion, we have quantified BODIPY staining and co-localization staining for *Plin2* and *Lamp2* (Fig.7 and Fig.S7a-d). We agree with the reviewer and have replaced “lipophagy” with “autophagy” in the revised manuscript.

13. How conserved are the sequences in the regions spliced in the mouse liver and mouse hepatocyte cells with human sequences? The authors should confirm that *Usp39* regulates this splicing event in a human hepatocyte line or primary hepatocyte (e.g. HepG2 or Huh7?) to support the splicing mechanism being the relevant one in humans to support downregulation of *Hsf1* in NASH patients.

Response: We appreciate the reviewer’s valuable comment. We compared the sequences of human and mouse *Hsf1* gene near the splice site of exon 6 and found the sequences was highly conserved (Figure 6 for reviewer) .

As suggested, human cell lines Huh7 and HepG2 were employed to analyze the splicing regulation of Usp39 on Hsf1. The semi-quantitative RT-PCR results showed that canonical isoform was significantly decreased while the NMD isoform was increased upon knockdown Usp39 in Huh7 and HepG2 cells (Revised Fig. S6f-i). Further immunoblotting showed that Usp39 knockdown decreased the protein level of Hsf1 in HepG2 and Huh7 cells (Revised Fig. S6o). These data verified that Usp39 regulates the alternative splicing and expression of Hsf1 in human hepatocytes.

References

- Schulze RJ, *et al.* Direct lysosome-based autophagy of lipid droplets in hepatocytes. *Proc Natl Acad Sci U S A* **117**, 32443-32452 (2020).
- Zhang Q, *et al.* Bub1 and Bub3 regulate metabolic adaptation via macrolipophagy in *Drosophila*. *Cell Rep* **42**, 112343 (2023).
- Fujiwara N, *et al.* Molecular signatures of long-term hepatocellular carcinoma risk in nonalcoholic fatty liver disease. *Sci Transl Med* **14**, eabo4474 (2022).
- Tang XH, *et al.* A retinoic acid receptor beta2 agonist attenuates transcriptome and

- metabolome changes underlying nonalcohol-associated fatty liver disease. *J Biol Chem* **297**, 101331 (2021).
5. Flores-Costa R, *et al.* Stimulation of soluble guanylate cyclase exerts antiinflammatory actions in the liver through a VASP/NF-kappaB/NLRP3 inflammasome circuit. *Proc Natl Acad Sci U S A* **117**, 28263-28274 (2020).
 6. Sun Z, *et al.* Hepatic Hdac3 promotes gluconeogenesis by repressing lipid synthesis and sequestration. *Nat Med* **18**, 934-942 (2012).
 7. He A, *et al.* Acetyl-CoA Derived from Hepatic Peroxisomal beta-Oxidation Inhibits Autophagy and Promotes Steatosis via mTORC1 Activation. *Mol Cell* **79**, 30-42 e34 (2020).
 8. Jiang P, Mizushima N. LC3- and p62-based biochemical methods for the analysis of autophagy progression in mammalian cells. *Methods* **75**, 13-18 (2015).
 9. Byun S, *et al.* Fasting-induced FGF21 signaling activates hepatic autophagy and lipid degradation via JMJD3 histone demethylase. *Nat Commun* **11**, 807 (2020).
 10. Desai S, *et al.* Heat shock factor 1 (HSF1) controls chemoresistance and autophagy through transcriptional regulation of autophagy-related protein 7 (ATG7). *J Biol Chem* **288**, 9165-9176 (2013).
 11. Zhang N, *et al.* HSF1 upregulates ATG4B expression and enhances epirubicin-induced protective autophagy in hepatocellular carcinoma cells. *Cancer Lett* **409**, 81-90 (2017).
 12. Kumsta C, Chang JT, Schmalz J, Hansen M. Hormetic heat stress and HSF-1 induce autophagy to improve survival and proteostasis in *C. elegans*. *Nat Commun* **8**, 14337 (2017).

REVIEWER COMMENTS

Reviewer #1 (Remarks to the Author):

Authors improved the manuscript considerably and responded to my comments accordingly.

Reviewer #2 (Remarks to the Author):

The Authors have responded to some of my previous comments. However, the following issues are still pending.

Figure S4 d. Food intake significantly decreased in Usp39-HKO mice at 5 weeks of age. This result has not been commented.

Fig. S4 m. In the Upper graph, the Seahorse profile of the siUSP39-OA cannot be found.

The Authors have not addressed my previous comment regarding the quantification and the statistical analysis of the western blots. Graphs with dots must be included. This refers to Western blots in the main and supplementary figures.

Reviewer #3 (Remarks to the Author):

The authors have responded adequately to my concerns. To enhance the value to readers I would recommend making the minor changes below to the manuscript prior to publication.

Fig.1g - decrease in Usp39 expression is not apparent in IHC. Please quantify and comment in the text.

Authors should acknowledge the second band for Usp39 and propose possible explanations in present text.

Fig.3m and Fig.5b - include at minimum enlarged versions of these including gene names for each RNA shown. Otherwise this is minimally useful to the reader as presented.

Figures for reviewers, such as quantification of p62 and LC3 mRNAs should be included in the paper. Also include Fig.1, 2 and 6 for reviewers in the paper as well at minimum. In Fig.6 for reviewers please highlight Motif1 for Hsf1 in the human and mouse sequences.

Response to reviewers' comments

We thank the reviewers again for your thoughtful comments and suggestions which help to further improve our manuscript. We have carefully reviewed the comments and have revised the manuscript accordingly. Our point-by-point responses to the reviewers' comments are included below. Text changes in the manuscript are highlighted in red.

Response to Reviewer #1 (Remarks to the Author)

----- **Reviewer comments:**

Authors improved the manuscript considerably and responded to my comments accordingly.

Response: We appreciate your positive comments on our revised manuscript.

Response to Reviewer #2 (Remarks to the Author)

----- **Reviewer comments:**

The Authors have responded to some of my previous comments. However, the following issues are still pending.

Response: We greatly appreciate your valuable comments and critiques. We have studied your comments point by point and revised the manuscript accordingly.

1. Figure S4 d. Food intake significantly decreased in Usp39-HKO mice at 5 weeks of age. This result has not been commented.

Response: Thank you for pointing this out, we have included the description in the revised manuscript on result part (lines 164-167).

2. Fig. S4 m. In the Upper graph, the Seahorse profile of the siUSP39-OA cannot be found.

Response: We thank the reviewer for bringing this to our attention. The panel is now presented as revised Fig. S6e.

3. The Authors have not addressed my previous comment regarding the quantification and the statistical analysis of the western blots. Graphs with dots must be included.

This refers to Western blots in the mail and supplementary figures.

Response: According to the reviewer's comment, we have now included quantification and statistical analysis of all western blot data, graphs with dots were included.

Response to Reviewer #3 (Remarks to the Author)

----- **Reviewer comments:**

The authors have responded adequately to my concerns. To enhance the value to readers I would recommend making the minor changes below to the manuscript prior to publication.

Response: We are pleased to hear that your concerns have been adequately addressed. We appreciate your suggestions to further improve our study.

1. Fig.1g - decrease in Usp39 expression is not apparent in IHC. Please quantify and comment in the text.

Response: Thank you for this suggestion. We have quantified the IHC signals and presented them in Fig.S1i and have included the description in the revised manuscript (lines 89-91).

2. Authors should acknowledge the second band for Usp39 and propose possible explanations in present text.

Response: We thank the reviewer for this suggestion. We have now added this point in the discussion on Page 4 (lines 84-88).

3. Fig.3m and Fig.5b - include at minimum enlarged versions of these including gene names for each RNA shown. Otherwise this is minimally useful to the reader as presented.

Response: Following the advice of the reviewer, we have included gene names on

Fig.3m and Fig.5b.

4. Figures for reviewers, such as quantification of p62 and LC3 mRNAs should be included in the paper. Also include Fig.1, 2 and 6 for reviewers in the paper as well at minimum. In Fig.6 for reviewers please highlight Motif1 for Hsf1 in the human and mouse sequences.

Response: Thank you for the suggestions. We have included figures for reviewers in the paper (Fig.S1e, f, Fig.S3a, b, Fig.S4d, Fig.S5n, o and Fig.S8a), and revised the manuscript accordingly.

REVIEWERS' COMMENTS

Reviewer #2 (Remarks to the Author):

The Authors have addressed my comments and the manuscript has substantially improved.

Response to reviewers' comments

We thank the reviewers again for the careful and insightful comments which lead to the improvement of our manuscript via the revision. See below the response to Review #2.

Response to Reviewer #2 (Remarks to the Author)

----- **Reviewer comments:**

The Authors have addressed my comments and the manuscript has substantially improved.

Response: We are pleased that we were able to adequately address the reviewer's concerns. We appreciate your suggestions to further improve our study.